# Atmospheric Disturbance Modelling for a Piloted Flight Simulation Study of Airplane Safety Envelope over Complex Terrain

**Xinying Liu** [1,2,*] **, Anna Abà** [2]**, Pierluigi Capone** [2]**, Leonardo Manfriani** [2] **and Yongling Fu** [1]

1 Laboratory of Aerospace Servo Actuation and Transmission, School of Mechanical Engineering and Automation, Beihang University, Xueyuan Road 37, Haidian District, Beijing 100191, China; fuyongling@buaa.edu.cn

2 Centre for Aviation, School of Engineering, Zurich University of Applied Sciences, Technikumstrasse 71, 8401 Winterthur, Switzerland; anna.aba@zhaw.ch (A.A.); pierluigi.capone@zhaw.ch (P.C.); leonardo.manfriani@zhaw.ch (L.M.)

\* Correspondence: xinying.liu@zhaw.ch; Tel.: +41-058-934-7849

**Abstract:** A concept of a new energy management system synthesizing meteorological and orographic influences on airplane safety envelope was developed and implemented at the ZHAW Centre for Aviation. A corresponding flight simulation environment was built in a Research and Didactics Simulator (ReDSim) to test the first implementation of the cockpit display system. A series of pilot-in-the-loop flight simulations were carried out with a group of pilots. A general aviation airplane model Piper PA-28 was modified for the study. The environment model in the ReDSim was modified to include a new ad hoc subsystem simulating atmospheric disturbance. In order to generate highly resolved wind fields in the ReDSim, a well-established large-eddy simulation model, the Parallelized Large-Eddy Simulation (PALM) framework, was used in the concept study, focusing on a small mountainous region in Switzerland, not far from Samedan. For a more realistic representation of specific meteorological situations, PALM was driven with boundary conditions extracted from the COSMO-1 reanalysis of MeteoSwiss. The essential variables (wind components, temperature, and pressure) were extracted from the PALM output and fed into the subsystem after interpolation to obtain the values at any instant and any aircraft position. Within this subsystem, it is also possible to generate statistical atmospheric turbulence based on the widely used Dryden turbulence model. The paper compares two ways of generating atmospheric turbulence, by combining the numerical method with the statistical model and introduces the flight test procedure with an emphasis on turbulence realism; it then presents the experiment results including a statistical assessment achieved by collecting pilot feedback on turbulence characteristics and turbulence/task combination.

**Keywords:** large-eddy simulation; atmospheric turbulence modelling; real-time flight simulation; flight safety; mountainous terrain



## 1. Introduction

According to the statistics included in the accident investigation reports published by the Swiss Transportation Safety Investigation Board (STSB), excessive accidents involving general aviation airplanes occur in complex mountainous terrain in Switzerland. The Centre for Aviation (ZAV) at the Zurich University of Applied Sciences (ZHAW) conducted a concept study for the development of a cockpit display system synthesizing meteorological and orographic influences on airplane safety envelope [1]. A corresponding flight simulation environment was built in the ZAV Research and Didactics Simulator (ReDSim) for piloted flight simulation study aiming to test the first implementation of the cockpit display system. An important prerequisite for the concept study is a realistic representation of the disturbances produced by the atmospheric motions and irregularities, particularly at low

levels over mountainous terrain. The environment model in the ReDSim was modified to include a new ad hoc subsystem simulating atmospheric disturbances that variously described as gusts, turbulence and wind shear.

Over the past few decades, different methods of modelling atmospheric disturbances were developed and implemented in flight simulators for the design, assessment and certification of aircraft as well as for human factor investigations such as evaluation of pilot's workload, upset prevention and recovery training. Before starting to introduce the methodology applied in this paper, it would be necessary to give a brief review of methods and philosophies related to atmospheric disturbance modelling. The review is by no means exhaustive, but draws the background of the presented work. In general, there are two opposite approaches to generate atmospheric disturbances in a flight simulator. One is using statistical methods for building stochastic turbulence fields, the other one is creating realistic turbulence by means of solving fluid motion equations numerically. The intent of each is quite different. In the case of the former, the total wind velocities consist of predefined mean values and modelled turbulent fluctuations. In the latter case the mean wind velocities are calculated by removing small scale perturbations.

Most statistical methods for simulating atmospheric turbulence can be summarized as a random procedure whereby a noise process with specified power spectral densities is filtered to obtain an output with certain statistical properties. This approach is also known as Monte Carlo turbulence simulation. The two most popular forms, suggested by von Kármán in [2] and Dryden in [3], are normally used to define the power spectral densities of the three turbulence velocity components. The spectral shapes suggested by Dryden can be exactly matched using linear filters and were more convenient for turbulence modelling in the fifties, while it was confirmed that von Kármán's form was the most accurate model for representing isotropic turbulence. According to the comparison conducted in [4] by using recorded gust velocity and meteorological data from flight tests, the differences between the two proposed approaches were quite small in the frequency range of interest in most flight simulation studies. Although Gaussian turbulence models are based on the assumption that turbulence is isotropic, homogeneous and frozen (refer to frozen turbulence hypothesis suggested by Taylor in [5]), the linearly filtered Gaussian white noise method was still commonly used in the applications in the fifties.

The development of non-Gaussian atmospheric turbulence models began in the sixties, in order to reproduce the non-Gaussian characteristics of real turbulence. Reeves' Model proposed originally in [6] was considered as a representative non-Gaussian model which was intended to be used in piloted flight simulators. Considering the patchy nature of turbulence, a product of independent Gaussian random signals was used to generate three turbulence velocity components. The amplitude probability density of the simulated velocity components was characterized by a modified Bessel function. It aimed to overcome the problem that Gaussian models resulted in an underestimate of the large velocities. The model was further modified and implemented in a motion-based flight simulator for collecting pilot experience [7]. In a later report [8] the method was developed to calculate aircraft response statistics, respectively, power spectral density, probability distribution and exceedance frequencies, requiring calculation of the eigenfunctions and eigenvalues of a certain kernel function by means of a digital computer and appropriate numerical methods. The statistical properties of the proposed model were compared with experimentally measured properties of low altitude atmospheric turbulences. The results showed good agreements of power spectral densities, probability distributions and exceedance frequencies, especially as far as high velocities were concerned.

Some other models around the non-Gaussian approach were proposed by Jones [9] and Van de Moesdijk [10,11] in the 1970s. In this decade, another significant advance in turbulence modelling was capturing the vertical nonhomogeneous character of turbulence. The model developed by Fichtl et al. [12] was able to incorporate interlevel coherence by adding together a series of filtered white noise sources and subsequently filtering the sum. In [13], Fichtl extended the multi-filter system and introduced a nondimensional

turbulence model accommodating variability of turbulence properties along an aerospace vehicle trajectory. The Dryden spectra were expressed as nondimensional functions of a nondimensional frequency $\Omega = Lf_\omega/V$ where $f_\omega$ is dimensional radian frequency, $L$ is the turbulence scale length and $V$ is the true airspeed of the aerospace vehicle. To obtain the desired characteristics, a band-limited white noise was passed through a forming filter parameterised by the nondimensional spectra. Furthermore, a statistical discrete gust model was first proposed for predicting the rate of large aircraft response amplitudes. As Jones explained in [14], the point of view underlying the statistical discrete gust model of turbulence is that in fact a turbulent flow field contains coherent structures which are more appropriately described in terms of spatial velocity distribution than by transforming to a spectral or frequency distribution. This review paper [14] summarized, inter alia, the development progress of discrete gust modelling and the corresponding simulation techniques up to the end of the 1970s. The approach to atmospheric disturbance modelling that accommodates stochastic turbulence and distributed discrete gusts was standardized in the military specification [15] and handbook [16] which are widely used in design and certification processes of fixed wing airplane. As one of the methods used in the current study for predicting atmospheric disturbances, the standardized model with Dryden spectra has been included in our flight simulation test environment. For a brief mathematical description of the model, the reader is referred to Section 2.

Further advances in atmospheric disturbance modelling arose from the need for enhanced realism in the simulation of flight through low altitude. The previously described approaches for simulating stationary and isotropic atmospheric turbulence were extended analytically, experimentally and numerically, with consideration of the nonstationary and anisotropic turbulence characteristics. The turbulence associated with realistic meteorological events in the atmospheric boundary layer, e.g., microburst, was of concern. The meteorological phenomenon was defined by Fujita in [17], as "a strong downdraft which induces an outburst of damaging wind at the surface" and subsequently quantified by Wilson et al. in [18], "as a strong downburst having surface with the distance over which the velocity difference occurs being between 0.4 and 4 km". The Joint Airport Weather Studies (JAWS) research project, which was jointly administrated from 1981 to 1984 by the US National Centre for Atmospheric Research (NCAR) and the University of Chicago, investigated the microburst events, having 2–10 km spatial and 2–10 min temporal scales, at Denver's Stapleton International Airport during the summer of 1982 [19]. As one of the achievements of the project, a comprehensive database was established for use in flight simulation and examining aircraft performance in wind shear conditions. Campbell [20] developed a spatial model containing the time histories of the three velocity components of the real wind shear interpolated from the JAWS data sets to predict mean values. The fluctuation terms were determined by adding frozen and isotropic turbulence varying over the three corresponding spatial directions. As an outlook of future work, this report addressed the requirement of further study on the effect of topography on microbursts and distributions of turbulence length scale, since the vertical relation for turbulence length scale was based on an simple empirical form. Another 1980s extension of the conventional stochastic approach to simulate low-altitude atmospheric turbulence was the modelling of anisotropic turbulence. The development of the method was presented in a sequence of reports [21–23].

Since the 1990s, research studies have focused on incorporating the interactions of atmospheric disturbances with terrain and aircraft, particularly for flight simulation of rotorcraft, as the effects of blade rotation must be taken into account. Initially in [24], McFarland extended the conventional method from the military specification [15] to helicopter blade elements. The model called Simulation of Rotor Blade Element Turbulence (SORBET) used a temporal and geometrical distribution algorithm to provide gust velocity components in real time at each station over the rotor plane. A reviewed version of the SORBET model was presented in [25].The new model named Finite Element Aircraft Simulation of Turbulence (FEAST) is applicable to conventional fixed-wing aircraft. In the FEAST

model, the conventional angular filters were replaced by using the geometrical separation of centres of pressure and a novel Gaussian distribution in spanwise, as developed from a two-dimensional vertical turbulence field. A three dimensional extension of the turbulence velocity distribution method was proposed by Ji et al. [26], with the consideration of the turbulence effect across the whole aircraft. For predicting discrete gusts and turbulence magnitudes, the FEAST model can use terrain related functions based on a simple model (canyon or ship deck) [25], or derived from computational fluid dynamics (CFD) simulations including specific orographic features of a visual database.

In the past over twenty years, besides methods based on experimental data such as reconstruction of wind fields from Light Detection And Ranging (LiDAR) measurements for wind energy applications [27–30], and developing atmospheric disturbance models of specific rotatory aircraft via system identification techniques using flight test data [31–33], research studies on turbulence modelling have more frequently adopted numerical approaches. This trend can be considered as a result of the significant advancement in computational capabilities. In aerodynamics and environment science, direct numerical simulations (DNS) and large eddy simulations (LES) have been increasingly applied to simulate complex and spatially developing flows. In order to solve partial differential equations, the boundary conditions must be imposed properly, in particular at inflow boundaries. In general, turbulence recycling and synthetic turbulence are at present the main two methods to specify instantaneous turbulent inlet boundary conditions. In the first case, the inflow data is obtained either from downstream outflow data of the same simulation or from upstream data of a precursor simulation. The second approach involves generating turbulence fluctuations by using random sequences. Kieting et al. [34] presented a comprehensive review of the methods to generate inflow boundary conditions for high fidelity flow simulations. As an efficient model requiring relatively low computation power, Jarrin et al. [35] introduced a Synthetic Eddy Method (SEM) which decomposes a turbulence flow field in a finite sum of eddies generating velocity fluctuations, using a suitable form suggested by Lund et al. [36] to describe the amplitude tensor of unscaled fluctuations. The method was essentially used for coupling between an upstream Reynolds Averaged Navier-Stokes (RANS) simulation and LES for channel flow. In a later report [37], the SEM approach was modified so that the input parameters of eddy generation only used the statistics available from the upstream RANS simulation. This method was extended by Kim et al. [38] for aeroacoustic investigation, and has been adapted recently for piloted flight simulation and handling qualities analysis [39]. A significant progress in inflow turbulence modelling for LES was made by Xie and Castro [40]. The method proposed in this paper adopted an exponent function based on two weight factors to impose correlations in time on random sequences. A modified version, in terms of inlet mass flux correction and satisfying divergence conditions for incompressible flow solver, was introduced in [41]. This method was originally applied in a numerical weather forecasting code to provide the dynamic large-scale inlet boundary conditions for the computations of small-scale urban canopy flows [40].

Our flight simulation study is a scenario-based investigation, which requires realistic meteorological conditions, possibly provided by a regional numerical mesoscale model. Moreover, the present study emphasizes the influences of meteorological conditions and complex mountainous terrain on the performance margin of a general aviation airplane, Piper PA-28, which has a wingspan and total length of less than 10 m. Hence, it is necessary to resolve the wind fields in the area of most interest with a grid spacing on the order of $\mathcal{O}(\sim 10 \text{ m})$. However, most regional mesoscale models, such as COSMO (Consortium for Small-scale Modelling), can provide wind fields with a horizontal spacing of $\mathcal{O}(\sim 1 \text{ km})$, which is considered as coarse for our application. In addition, the wind data derived from such mesoscale models might present a lack of fluctuations, as the small-scale turbulences with scale length of less then several kilometers are completely parameterized. Considering the above aspects, a micro-scale model synoptically evolving conditions in high resolution, is desirable for our study. A well-established model named Parallelized Large-Eddy

Simulation Model (PALM) meets the requirements of our application. The model system has been mainly developed by the PALM group at the Institute of Meteorology and Climatology at Leibniz University Hanover, Germany. As one of the first parallelized LES models for atmospheric turbulence study, the original code [42] which is the core of the current PALM version has been continually improved, extended and further developed, increasingly over the last ten years. A series of submodels are embedded in the recent versions. The PALM model system has been widely applied for simulations of urban climate, atmospheric and ocean boundary layers. As a comprehensive version, PALM 4.0 was released by Maronga et al. [43] in 2015. A master thesis [44] from the development team presented a case study on comparing wind fields derived from PALM with the ones generated by flight simulator software FlightGear, under hypothetically predefined flight condition. We followed the similar approach but conducted a plioted simulation study under a realistic meteorological condition. In [45], the development team updated the latest modifications in PALM version 6.0, which was used for our simulations. PALM provides a mesoscale nesting interface combined with a synthetic turbulence generator using the previously mentioned Xie and Castro's method [40]. This feature allows to force COSMO data onto the lateral and top domain boundaries of PALM simulation. A recently published paper [46] describes explicitly the implementation of the mesoscale-nesting interface in PALM and compares the turbulence statistics generated by PALM with the mesoscale model COSMO.

The present paper is structured as follows. We describe the discrete form of the Dryden wind turbulence model in Section 2 and present an overview of the PALM model system core and relevant features, including governing equations, turbulence closure, numerics and boundary conditions in Section 3. We then introduce the implementation in our flight simulator and test program of the flight simulation campaign in Section 4. In Section 5, we evaluate the numerical results from PALM and demonstrate a comparison of two combinations of the numerical simulation and Dryden wind turbulence model for simulating atmosphere disturbances, followed by a handling quality analysis derived from the piloted flight simulations. Finally, we summarize our work and give an outlook of future potential applications of the present method and implementation in Section 6.

## 2. Stochastic Method of Atmospheric Turbulence Modelling

The Dryden wind turbulence Simulink toolbox was integrated in the flight simulation test environment to calculate turbulence velocities. The block was implemented based on the mathematical representation in the Military Specifications [15]. The Dryden form of turbulence velocities in the three spatial directions is given as follows:

$$\Phi_{u_g}(\omega) = \frac{2\sigma_u^2 L_u}{\pi V} \cdot \frac{1}{1 + \left(L_u \frac{\omega}{V}\right)^2}, \tag{1}$$

$$\Phi_{v_g}(\omega) = \frac{\sigma_v^2 L_v}{\pi V} \cdot \frac{1 + 3\left(L_v \frac{\omega}{V}\right)^2}{\left[1 + \left(L_v \frac{\omega}{V}\right)^2\right]^2}, \tag{2}$$

$$\Phi_{w_g}(\omega) = \frac{\sigma_w^2 L_w}{\pi V} \cdot \frac{1 + 3\left(L_w \frac{\omega}{V}\right)^2}{\left[1 + \left(L_w \frac{\omega}{V}\right)^2\right]^2}. \tag{3}$$

Here, the original symbols in [15] are used. $\omega$ is the temporal frequency and $V$ is the airplane true airspeed. $u_g, v_g, w_g$ are the longitudinal, lateral and vertical turbulence velocity components, $L_u, L_v, L_w$ represent the turbulence scale length and $\sigma_u, \sigma_v, \sigma_w$ are the turbulence intensities. According to the military references, turbulence intensities and turbulence lengths are functions of altitude $h$, divided into three regions:

- low-altitude $h < 1000$ ft, where

$$L_w = h,$$

$$L_u = L_v = \frac{h}{(0.177 + 0.000823h)^{1.2}}, \tag{4}$$

$$\sigma_w = 0.1W_{20},$$

$$\frac{\sigma_u}{\sigma_w} = \frac{\sigma_v}{\sigma_w} = \frac{1}{(0.177 + 0.000823h)^{0.4}}, \tag{5}$$

with the typical values of the wind speed $W_{20}$ being 15 knots for light turbulence, 30 knots for moderate turbulence, 45 knots for severe turbulence [47];

- medium/high altitudes $h > 2000$ ft, where the turbulence is assumed as isotropic with $L_u = L_v = L_w = 1750$ ft and $\sigma_u = \sigma_v = \sigma_w$ determined from a lookup table referred to Figure 7 in [15];

- between low and medium/high altitudes $1000$ ft $< h < 2000$ ft, where turbulence quantities are determined by linearly interpolating between the values from the low altitude model at 1000 ft and the values from the high altitude model at 2000 ft [47].

According to [15], the spectra of the angular velocity disturbances due to turbulence are given by:

$$\Phi_{p_g}(\omega) = \frac{\sigma_w^2}{VL_w} \cdot \frac{0.8 \left(\frac{\pi L_w}{4b}\right)^{1/3}}{1 + \left(\frac{4b\omega}{\pi V}\right)^2}, \tag{6}$$

$$\Phi_{r_g}(\omega) = \frac{\left(\frac{\omega}{V}\right)^2}{1 + \left(\frac{3b\omega}{\pi V}\right)^2} \cdot \Phi_{v_g}(\omega), \tag{7}$$

$$\Phi_{q_g}(\omega) = \frac{\left(\frac{\omega}{V}\right)^2}{1 + \left(\frac{4b\omega}{\pi V}\right)^2} \cdot \Phi_{w_g}(\omega), \tag{8}$$

with the airplane wingspan $b$ and the following definitions of angular velocities based on a Taylor series approximation:

$$q_g = \frac{\partial w_g}{\partial x}, \quad p_g = \frac{\partial w_g}{\partial y}, \quad r_g = -\frac{\partial v_g}{\partial x}. \tag{9}$$

To generate a time series of turbulence velocities and angular velocities with Dryden spectral characteristics, a band-limited white noise signal is passed through forming filters. When integrating Equations (7) and (8) in the frequency domain, it can be seen that the spectral densities the variances $\Phi_{r_g}^2$ and $\Phi_{q_g}^2$ become infinite. Physically, this means that the air particles of turbulence motion would be infinite which is not possible in real processes. A detailed discussion about this mathematical drawback and its effect on further calculation of aircraft motion was presented in [48,49]. To overcome this problem, the original spectral are factored prior to filtering. The transfer functions of Dryden spectral densities can be written in Laplace form as follows:

$$G_{u_g}(s) = \sigma_u \sqrt{\frac{L_u}{\pi V}} \cdot \frac{1}{1 + \frac{L_u}{V}s}, \tag{10}$$

$$G_{v_g}(s) = \sigma_v \sqrt{\frac{L_v}{\pi V}} \cdot \frac{1 + \frac{\sqrt{3}L_v}{V}s}{\left(1 + \frac{L_v}{V}s\right)^2}, \tag{11}$$

$$G_{w_g}(s) = \sigma_w \sqrt{\frac{L_w}{\pi V}} \cdot \frac{1 + \frac{\sqrt{3}L_w}{V}s}{\left(1 + \frac{L_w}{V}s\right)^2}, \tag{12}$$

$$G_{p_g}(s) = \sigma_w \sqrt{\frac{0.8}{V}} \cdot \frac{\left(\frac{\pi}{4b}\right)^{1/6}}{L_w^{1/3}\left(1 + \left(\frac{4b}{\pi V}\right)s\right)}, \tag{13}$$

$$G_{r_g}(s) = \frac{\frac{s}{V}}{\left(1 + \left(\frac{3b}{\pi V}\right)s\right)} \cdot G_v(s), \tag{14}$$

$$G_{q_g}(s) = \frac{\frac{s}{V}}{\left(1 + \left(\frac{4b}{\pi V}\right)s\right)} \cdot G_w(s), \tag{15}$$

In currently widely used applications for flight simulators, the original spectra of $\Phi_{p_g}$ have been approximated. One common approximation was published by Yeager [50] and tested in a nonlinear six-degree-of-freedom flight simulator. In Yeager's implementation, the spectral of pitch velocity is written as

$$\Phi_{p_g}(\omega) = \frac{2\sigma_u^2 L_p}{\pi V} \cdot \frac{1}{1 + \left(L_u \frac{\omega}{V}\right)^2}, \tag{16}$$

with the corresponding turbulence scale length $L_p$ and turbulence intensity $\sigma_p$ being

$$L_p = \frac{\sqrt{L_w b}}{2.6}, \tag{17}$$

$$\sigma_p = \frac{1.9}{\sqrt{L_w b}}\sigma_v. \tag{18}$$

The finite difference equations of the filters Equations (10)–(15) read as

$$u_g^n = \left(1 - \frac{V}{L_u}T\right)u_g^{n-1} + \sigma_u\sqrt{2\frac{V}{L_u}T}N_{u_g}^{n-1}, \tag{19}$$

$$v_g^n = \left(1 - \frac{V}{L_u}T\right)v_g^{n-1} + \sigma_v\sqrt{2\frac{V}{L_u}T}N_{v_g}^{n-1}, \tag{20}$$

$$w_g^n = \left(1 - \frac{V}{L_u}T\right)w_g^{n-1} + \sigma_w\sqrt{2\frac{V}{L_u}T}N_{w_g}^{n-1}, \tag{21}$$

$$p_g^n = \left(1 - \frac{2.6}{\sqrt{L_w b}}T\right)p_g^{n-1} + \sigma_w\left(\sqrt{2\frac{2.6}{\sqrt{L_w b}}T}\right)\left(\frac{0.95}{\sqrt[3]{2L_w b^2}}\right)N_{p_g}^{n-1}, \tag{22}$$

$$r_g^n = \left(1 - \frac{\pi V}{3b}T\right)r_g^{n-1} + \frac{\pi}{3b}\left(v_g^n - v_g^{n-1}\right), \tag{23}$$

$$q_g^n = \left(1 - \frac{\pi V}{4b}T\right)q_g^{n-1} + \frac{\pi}{4b}\left(w_g^n - w_g^{n-1}\right). \tag{24}$$

with $N$ being the input signal and the subscript n denoting the timestep.

## 3. Numerical Procedure

PALM model system was used in our application to generate highly resolved wind fields for simulating atmospheric disturbances in the flight simulator. PALM is a pressure-based and finite-difference flow solver, which solves non-hydrostatic, filtered, incompressible Navier–Stokes equations. By default, PALM estimates seven quantities: the three

velocity components $u, v, w$ on a Cartesian grid, potential temperature $\theta$, sub-grid scale (SGS) turbulence kinetic energy (TKE) $e$, water vapor mixing ratio $q_v$ and possibly a passive scalar $s$. To ensure a self-contained paper, a brief description of the PALM model system core including some LES fundamentals and relevant features, boundary conditions and numerical methods used in our study is presented in this section. Table 1 presents the general parameters in PALM model system. For a detailed description of all PALM components and the code structure, the readers are referred to the publications [43,45] and PALM home website [51].

**Table 1.** List of general parameters in PALM model system.

| Symbol | Value | Description |
|---|---|---|
| $c_{\mathrm{m}}$ | 0.1 | SGS model constant |
| $c_{\mathrm{p}}$ | 1005 J kg$^{-1}$ K$^{-1}$ | Heat capacity of dry air at constant pressure |
| $g$ | 9.81 ms$^{-2}$ | Gravitational acceleration |
| $p_0$ | 1000 hPa | Reference pressure |
| $R_{\mathrm{d}}$ | 287 J kg$^{-1}$ K$^{-1}$ | Specific gas constant for dry air |
| $R_{\mathrm{v}}$ | 461.51 J kg$^{-1}$ K$^{-1}$ | Specific gas constant for water vapor |
| $\kappa$ | 0.4 | Von Kármán constant |
| $\Omega$ | $\Omega = 0.729 \times 10^{-4}$ rads$^{-1}$ | Angular velocity of the Earth |

### 3.1. Governing Equations

PALM solves in principle incompressible Navier-Stokes equations but allows for density variations with height, as presented in an anelastic form in [45]. As we focused on simulating the lowest several kilometers of troposphere, here we still use the Boussinesq-approximated form to introduce the governing equations, in which the density $\rho$ is treated as a constant value and drops out of the most terms. A general form of the equation for conservation of momentum is given as follows,

$$\frac{\partial u_i}{\partial t} = -\frac{\partial u_i u_j}{\partial x_j} - \epsilon_{ijk} f_j u_k + \varepsilon_{i3j} f_3 u_{\mathrm{g},j} - \frac{1}{\rho_0}\frac{\partial p^*}{\partial x_i} + \nu \frac{\partial^2 u_i}{\partial x_j^2} + g\frac{\theta_{\mathrm{v}} - \theta_{\mathrm{v,ref}}}{\theta_{\mathrm{v,ref}}}\delta_{i3}. \tag{25}$$

Here, $i, j, k \in \{1, 2, 3\}$. $u_i$ are the velocity components ($u_1 = u, u_2 = v, u_3 = w$) with Cartesian coordinates $x_i(x_1 = x, x_2 = y, x_3 = z)$, $t$ is time. To include Coriolis force contribution, the velocity component $u_k$ is multiplied with the Coriolis coefficient $f_i = (0, 2\Omega\cos(\phi), 2\Omega\sin(\phi))$ with $\Omega = 0.729 \times 10^{-4}$ rads$^{-1}$ being the Earth's angular velocity and $\phi$ being the geographical latitude, $\varepsilon$ is the Levi-Civity symbol. Furthermore, the synoptic-scale pressure gradient is added to the momentum equation by multiplying the geostrophic wind speed components $u_{g,j}$ with the corresponding vector parameter. The subscript 0 indicates the surface value, $p^*$ is the perturbation pressure. The next term in Equation (25) represents the molecular friction with the kinematic viscosity $\nu$. The buoyancy term is expressed with the gravitational acceleration $g = 9.81$ ms$^{-2}$, the Kronecker delta $\delta$, and the virtual potential temperature $\theta_{\mathrm{v}}$. The reference state $\theta_{\mathrm{v,ref}}$ can be set to be the horizontal average $\langle\theta_v\rangle$ used in [43], the initial state, or a fixed reference value. The potential temperature is defined as

$$\theta = T/\Pi, \tag{26}$$

with the absolute temperature $T$ and the Exner function:

$$\Pi = \left(\frac{p}{p_0}\right)^{R_{\mathrm{d}}/c_{\mathrm{p}}}. \tag{27}$$

Here, $p$ is the hydrostatic air pressure, $p_0 = 1000$ hPa is a reference pressure, $R_d = 287$ J kg$^{-1}$ K$^{-1}$ is the specific gas constant for dry air, and $c_p = 1005$ J kg$^{-1}$ K$^{-1}$ is the specific heat of dry air at constant pressure. The virtual potential temperature is defined as

$$\theta_v = \theta \left[ 1 + \left( \frac{R_v}{R_d} - 1 \right) q_v - q_l \right], \tag{28}$$

with the specific gas constant for water vapor $R_v = 461.51$ J kg$^{-1}$ K$^{-1}$, and the liquid water mixing ratio $q_l$. Two SGS models were used for benchmaking purpose in the present paper.

PALM adopts a spatial scale separation approach developed by Schumann [52] to implicitly filter the variables in the governing equations. When applying the spatial filter to Equation (25), the filtered nolinear advection term $\overline{u_i u_j}$ can be written as

$$\overline{u_i u_j} = \overline{\bar{u}_i \bar{u}_j} + \underbrace{\overline{u_i'' + u_j''}}_{R_{ij}} + \underbrace{\overline{\bar{u}_i u_j''} + \overline{u_j'' \bar{u}_i}}_{C_{ij}} \tag{29}$$

with the Reynolds-stress $R_{ij}$ and the cross-stress $C_{ij}$ which vanishes in RANS model. The overbar indicates filtered quantities in general. The double prime indicates SGS variables. Furthermore, the first term of the right-hand side can be formulated by using Leonard-stress $L_{ij}$

$$\overline{\bar{u}_i \bar{u}_j} = \bar{u}_i \bar{u}_j + \underbrace{\left( \overline{\bar{u}_i \bar{u}_j} - \bar{u}_i \bar{u}_j \right)}_{L_{ij}} \tag{30}$$

Considering Equations (29) and (30), a filtered form of Equation (25) reads as

$$\begin{aligned}
\frac{\partial \bar{u}_i}{\partial t} = & -\frac{\partial \bar{u}_i \bar{u}_j}{\partial x_j} - \varepsilon_{ijk} f_j \bar{u}_k + \varepsilon_{i3j} f_3 u_{g,j} - \frac{\partial \pi^*}{\partial x_i} \\
& + g \frac{\bar{\theta}_v - \theta_{v,\text{ref}}}{\theta_{v,\text{ref}}} \delta_{i3} - \frac{\partial \tau_{ij}^d}{\partial x_j},
\end{aligned} \tag{31}$$

with the deviatoric subgrid stress $\tau_{ij}^d$ subtracting the normal stress term from the total subgrid stress $\tau_{ij}$:

$$\begin{aligned}
\tau_{ij}^d = & \underbrace{L_{ij} + C_{ij} + R_{ij}}_{\tau_{ij}} - \frac{1}{3} \tau_{kk} \delta_{ij}, \\
= & \left( \overline{u_i'' u_j''} - \frac{2}{3} e \delta_{ij} \right),
\end{aligned} \tag{32}$$

with the modified perturbation:

$$\pi^* = p^* + \frac{2}{3} \rho_0 e. \tag{33}$$

with the SGS-TKE:

$$e = \frac{1}{2} \overline{u_i'' u_i''}. \tag{34}$$

Analogously, the filtered equations for conservation of mass, thermal internal energy, moisture, and another arbitrary passive scalar can be written as

$$\frac{\partial \bar{u}_j}{\partial x_j} = 0. \tag{35}$$

$$\frac{\partial \bar{\theta}}{\partial t} = -\frac{\partial \bar{u}_j \bar{\theta}}{\partial x_j} - \frac{\partial}{\partial x_j} \left( \overline{u_j'' \theta''} \right) - \frac{l_v}{c_p \Pi} \chi_{q_v}, \tag{36}$$

$$\frac{\partial \overline{q}_{\mathrm{v}}}{\partial t} = -\frac{\partial \overline{u}_j \overline{q}_{\mathrm{v}}}{\partial x_j} - \frac{\partial}{\partial x_j}\left(\overline{u_j'' q_v''}\right) + \chi_{q_v}, \tag{37}$$

$$\frac{\partial \overline{s}}{\partial t} = -\frac{\partial \overline{u}_j \overline{s}}{\partial x_j} - \frac{\partial}{\partial x_j}\left(\overline{u_j'' s''}\right) + \chi_s, \tag{38}$$

with $\chi_{q_v}$ and $\chi_s$ being the source/sink terms of $q_v$ and $s$, and the specific latent heat of vaporization $l_{\mathrm{v}} = 2.5 \times 10^6 \ \mathrm{J \ kg^{-1}}$.

*3.2. Turbulence Closure*

By default, PALM model is operated as LES model using a subgrid-scale model to parameterize the effect of the small-scale turbulence which has a smaller size than the grid size. Taking the idea from RANS modelling, PALM uses an eddy viscosity type SGS model to parameterize the four SGS convariance terms: $\overline{u_i'' u_j''}, \overline{u_j'' \theta''}, \overline{u_j'' q_v''}, \overline{u_j'' s''}$ in Equations (31) and (36)–(38).

3.2.1. Deardorff Subgrid-Scale Model

The default SGS model in PALM follows a 1.5-order closure approach suggested by Deardorff [53]. The modified version of [54,55] was implemented in PALM. The closure assumes that the SGS terms presenting energy transport by SGS eddies are proportional to the local gradients of the considered variables and read as

$$\tau_{ij}^d = \overline{u_i'' u_j''} - \frac{2}{3} e \delta_{ij} = -2K_{\mathrm{m}} \overline{S}_{ij}, \tag{39}$$

$$\overline{u_i'' \theta''} = -K_{\mathrm{h}} \frac{\partial \overline{\theta}}{\partial x_i}, \tag{40}$$

$$\overline{u_i'' q_v''} = -K_{\mathrm{h}} \frac{\partial \overline{q}_{\mathrm{v}}}{\partial x_i}, \tag{41}$$

$$\overline{u_i'' s''} = -K_{\mathrm{h}} \frac{\partial \overline{s}}{\partial x_i}, \tag{42}$$

with the definition of the rate of strain tensor

$$\overline{S}_{ij} = 0.5\left(\frac{\partial \overline{u}_i}{\partial x_j} + \frac{\partial \overline{u}_j}{\partial x_i}\right), \tag{43}$$

with the local SGS eddy diffusivities of momentum $K_{\mathrm{m}}$ and heat $K_{\mathrm{h}}$ defined as

$$K_{\mathrm{m}} = c_{\mathrm{m}} l \sqrt{e}, \tag{44}$$

$$K_{\mathrm{h}} = \left(1 + \frac{2l}{\Delta}\right) K_{\mathrm{m}}. \tag{45}$$

Here, $c_{\mathrm{m}}$ is the model constant, the cutoff length $\Delta$ represents the largest size of the unresolved scales. $\Delta$ is determined by distance from the ground $z$ and the grid spacings $\Delta x, \Delta y, \Delta z$ in $x, y$ and $z$ direction can be written as

$$\Delta = \min\left(1.8z, (\Delta x \Delta y \Delta z)^{\frac{1}{3}}\right). \tag{46}$$

The SGS mixting length $l$ is function of height $z$, grid spacing and and calculated as

$$l = \begin{cases} \min\left(\Delta, 0.76\sqrt{e}\left(\frac{g}{\theta_{\mathrm{v,ref}}}\frac{\partial \theta_{\mathrm{v}}}{\partial z}\right)^{-\frac{1}{2}}\right) & \text{for } \frac{\partial \theta_{\mathrm{v}}}{\partial z} > 0, \\ \Delta & \text{for } \frac{\partial \theta_{\mathrm{v}}}{\partial z} \leq 0. \end{cases} \tag{47}$$

Moreover, the closure includes a prognostic equation for the SGS-TKE as follows

$$\frac{\partial e}{\partial t} = -u_j \frac{\partial e}{\partial x_j} - \left( \overline{u_i'' u_j''} \right) \frac{\partial u_i}{\partial x_j} + \frac{g}{\theta_{\mathrm{v,ref}}} \overline{u_3'' \theta_{\mathrm{v}}''} - \frac{\partial}{\partial x_j} \left[ \overline{u_j'' \left( e + \frac{p''}{\rho_0} \right)} \right] - \epsilon. \tag{48}$$

The pressure term in the above equation is defined as

$$\left[ \overline{u_i'' \left( e + \frac{p''}{\rho_0} \right)} \right] = -2K_{\mathrm{m}} \frac{\partial e}{\partial x_j}, \tag{49}$$

and the SGS dissipation $\epsilon$ within a grid volume can be written as

$$\epsilon = \left( 0.19 + 0.74 \frac{l}{\Delta} \right) \frac{e^{\frac{3}{2}}}{l}. \tag{50}$$

According to the definition in Equation (28) and the Stull's description [56], the vertical SGS buoyancy flux reads as

$$\overline{u_k'' \theta_{\mathrm{v}}''} = K_1 \cdot \overline{u_k'' \theta''} + K_2 \cdot \overline{u_k'' q_{\mathrm{v}}''} - \theta \cdot \overline{u_k'' q_1''}, \tag{51}$$

with

$$K_1 = 1 + \left( \frac{R_{\mathrm{v}}}{R_{\mathrm{d}}} - 1 \right) \overline{q_{\mathrm{v}}} - \overline{q_1}, \tag{52}$$

$$K_2 = \left( \frac{R_{\mathrm{v}}}{R_{\mathrm{d}}} - 1 \right) \overline{\theta}, \tag{53}$$

and the vertical SGS flux of liquid water, parametrized as:

$$\overline{w'' q''} = -K_{\mathrm{h}} \frac{\partial q_1}{\partial z}. \tag{54}$$

### 3.2.2. Dynamic Subgrid-Scale Model

In Deardorff's model, $c_m$ in Equation (44) is considered as constant which actually varies flow types. The model with constant $c_m$ overestimates the velocity shear in the near wall region, particularly if the grid resolution is not sufficiently fine. To overcome these deficiencies, many research were carried out in the 1980s to develop dynamic SGS models whereby $c_s$ varies spatially and temporally. By describing the dynamic SGS model implemented in PALM, the $c_m$ is written as $c_*$ and the $l$ in Equation (44) is replaced by the symbol $\Delta_{\mathrm{max}}$. As a pioneer study on developing dynamic SGS, Germano et al. [57] introduced a test filter $\Delta_T = 2\Delta_{\mathrm{max}}$ and the Germano identity $\mathcal{L}_{ij}$ defined as

$$\mathcal{L}_{ij} = T_{ij} - \widehat{\tau}_{ij} = \widehat{\overline{u_i u_j}} - \widehat{\overline{u}}_i \widehat{\overline{u}}_j, \tag{55}$$

with the hat denoting filter operation with the filter length $\Delta_T$ and the subgrid stress on the test filter $T_{ij}$ calculated as

$$T_{ij} = \widehat{\overline{u_i u_j}} - \widehat{\overline{u}}_i \widehat{\overline{u}}_j. \tag{56}$$

While $\mathcal{L}_{ij}$ can be resolved, $T_{ij}$ is still unknown. To eliminate the unknown $T_{ij}$ for calculating $\widehat{\tau}_{ij}$, Heinz [58] used the deviatoric components of the Germano identity written as $\mathcal{L}_{ij}^d$ and the following eddy viscosity type assumption:

$$T_{ij}^d = -2\nu_t^T \widehat{S}_{ij}^d, \tag{57}$$

with the subtest-scale viscosity $v_t^T$ calculated as

$$v_t^T = c_* \left( \Delta^T \right)^2 \left| \widehat{\widetilde{S}}^d \right|,$$ (58)

with the filtered characteristic strain rate $\left| \widehat{\widetilde{S}}^d \right|$ given by

$$\left| \widehat{\widetilde{S}}^d \right| = \left( 2 \widehat{\widetilde{S}}_{mn} \widehat{\widetilde{S}}_{nm} \right)^{1/2}.$$ (59)

Based on Equations (57)–(59), $c_*$ can be written as

$$c_* = -\frac{\mathcal{L}_{ij}^d \widehat{\widetilde{S}}_{ji}}{2 v_t^{T} \widehat{\widetilde{S}}_{mn} \widehat{\widetilde{S}}_{nm}}.$$ (60)

To ensure the stability, Mokhatarpoor and Heinz [59] suggested a dynamic bounds that keep the values $c_*$ in the ranges

$$|c_*| \le \frac{23}{24\sqrt{3}} \frac{\sqrt{e}}{\Delta \sqrt{\overline{S}_{ij} \overline{S}_{ji}}}.$$ (61)

The dynamic model was used as default model in the present study.

*3.3. Discretization and Numerics*

3.3.1. Numerical Grid

In PALM, the calculation domain is discretized by using finite differences and equidistant horizontal grid spacings $\Delta x, \Delta y$. By default, the model uses constant grid spacing $\Delta z$ along the z-direction, but it can be stretched in the vertical direction to save computational time in the free atmosphere. In our application of simulating general airplane flights over a mountainous region, we focused on a limited height range from about 2000 m to less than 5000 m above mean sea level (AMSL). The existing computing resources allowed us to conduct the simulations with fine grid spacings in vertical direction. Hence, we kept the default setting with a constant grid spacing. PALM uses a C-grid topology called Arakawa staggered C-grid defined in [60,61]. By definition of the Arakawa staggered C-grid, all scalar variables $s$, e.g., the perturbation pressure $p^*$, SGS-TKE $e$, SGS eddy diffusivities of momentum $K_m$ and heat $K_h$ etc., locate at the cell centres, while the vector variables, respectively the wind components $u, v, w$ are shifted by half of the grid spacing to the face centres (see Figure 1). In the vertical direction, the first $w$-component $w(k = 0)$ indicates the bottom boundary at the surface height and the last $w$-component $w(k = nz)$ defines the top boundary. The first grid point for scalars $s$ and horizontal wind components $u, v$ locates half grid spacing $\Delta z/2$ under the surface. However, this point is just declared in the array but practically not required for the calculation. At the top boundary, the last grid point $nz + 1$ of $s, u, v$ is $\Delta z/2$ above $w(k = nz)$.

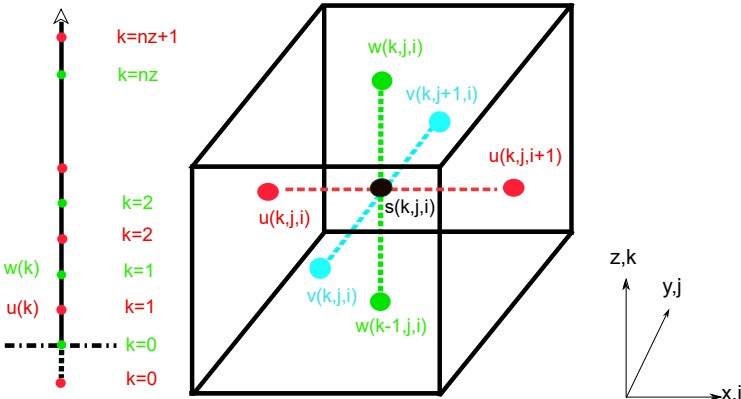

**Figure 1.** The Arakawa staggered C-grid.

### 3.3.2. Numerical Scheme

For discretisizing the advective terms $\frac{\partial \psi_i}{\partial t}$ in the governing equations, the following semidiscrete form is used in PALM:

$$\frac{\partial \psi_i}{\partial t} = -\frac{F_{i+\frac{1}{2}}(u\psi) - F_{i-\frac{1}{2}}(u\psi)}{\Delta x} \tag{62}$$

with the $F_{i\pm\frac{1}{2}} = u_{i\pm\frac{1}{2}}\psi_{i\pm 1}$ referring to the fluxes staggered half a grid length related to the advected quantity.

Using a 5th order upwind discretization suggested by Wicker and Skamarock [62], which is consisting of centered non dissipative 6th oder flux and an artificial added numerical dissipation term, the flux can be written as

$$F_{i-\frac{1}{2}}^{\text{6th}} = \underbrace{\frac{u_{i-\frac{1}{2}}}{60}[37(\psi_i + \psi_{i-1}) - 8(\psi_{i+1} + \psi_{i-2}) + (\psi_{i+2} + \psi_{i-3})]}_{\text{6th finite difference approximation}}$$

$$\underbrace{-\frac{\left|u_{i-\frac{1}{2}}\right|}{60}[10(\psi_i - \psi_{i-1}) - 5(\psi_{i+1} - \psi_{i-2}) + (\psi_{i+2} - \psi_{i-3})]}_{\text{added numerical dissipation term of 5th order}} \tag{63}$$

The spatial scheme is combined with a 3rd order low-storage Runge-Kutta scheme recommended by Williamson [63]. The scheme reads as

$$\psi(t + \Delta t) = \psi(t) + \frac{1}{30}\Delta t(5k_1 + 9k_2 + 16k_3)$$
$$k_1 = F(\psi(t))$$
$$k_2 = F\left(\psi(t) + \frac{1}{3}\Delta t \cdot k_1\right) \tag{64}$$
$$k_3 = F\left(\psi(t) - \frac{3}{16}\Delta t \cdot k_1 + \frac{15}{16}\Delta t \cdot k_2\right),$$

with the timestep $\Delta t$.

### 3.3.3. Pressure Solver

By solving the governing equations in the Boussinesq-approximated form and using the previously described numerical scheme, incompressibility is required. PALM adopts a predictor-corrector method. As the first step, the momentum equation Equation (30) is solved without the perturbation pressure term $-\frac{\partial \pi^*}{\partial x_i}$. Consequently, a divergence velocity

field in the next step $t + \Delta t$ is calculated as follows, with the subscript prov denoting provisional velocity field and gradient

$$\overline{u}_{i_{\text{prov}}}^{t+\Delta t} = \overline{u}_i^t + \Delta t \left( \frac{\partial \overline{u}_i^t}{\partial t} \right)_{prov}. \tag{65}$$

In the second step, all remaining divergences are assigned to the perturbation pressure term $-\frac{\partial \overline{\pi^*}}{\partial x_i}$. Subsequently, the corrected velocity can be written as

$$\overline{u}_i^{t+\Delta t} = \overline{u}_{i_{\text{prov}}}^{t+\Delta t} + \Delta t \left( -\frac{\partial \overline{\pi^*}^t}{\partial x_i} \right). \tag{66}$$

Thirdly, to obtain a corrected velocity free of divergence, the first spatial derivative should be defined as 0. The definition denoted by the symbol $\overset{!}{=}$ leads to solve a Possion equation for the perturbation pressure $\overline{\pi^*}$

$$\frac{\partial}{\partial x_i} \overline{u}_i^{t+\Delta t} = \underbrace{\frac{\partial}{\partial x_i} \left( \overline{u}_{i_{\text{prov}}}^{t+\Delta t} - \Delta t \frac{\partial \overline{\pi^*}^t}{\partial x_i} \right)}_{\text{Possion equation}} \overset{!}{=} 0. \tag{67}$$

PALM provides two general categories to solve pressure: (i) Fast Fourier Transform (FFT) method, (ii) multigrid method. Various FFT methods were implemented in PALM model system and external FFT libraries can be used for cases of cyclic lateral boundary conditions. In our case of non-cyclic boundary conditions, we used the second method implemented PALM, an iterative multigrid scheme described in [64]. The scheme is based on Gauss-Seidel method defined as follows

$$p_{i,j}^{k+1} = \frac{1}{4} \cdot \left( p_{i-1,j}^k + p_{i+1,j}^k + p_{i,j-1}^k + p_{i,j+1}^k - \Delta x^2 f(i,j,k) \right). \tag{68}$$

Here, $k$ is the iteration step. The local scheme requiring only four neighbouring points at each iteration, converges slowly on the one hand, and can reduce local divergence on grids with different grid spacing on the other hand.

*3.4. Boundary Conditions*

3.4.1. Topography and Surface Boundary Condition

In order to conduct realistic simulations of atmospheric turbulence and its propagation, topography information must be specified properly, particularly in our study related to orographically complex mountainous terrain. PALM's topography implementation is based on the mask method introduced in [65]. The method resolves solid obstacles, such as buildings and orography, and treats the grid cell either as 100% fluid cell or 100% solid cell. PALM's topography features have been frequently revised and further developed, specially for urban climate research. In the PALM version 6.0, the topography implementation allows to simulate overhanging structures, e.g., bridges, tunnels, and the surface boundary condition is coupled with various sub-models, such as gas-phase chemistry, aerosol physics, wind turbine model, radiation processes, urban surface model including pavements and so on [45]. For the present study, we focused essentially on convective boundary layer and used the "classical" implementation in PALM, which reduces the 3-D obstacle dimension to a 2.5-D topography format. The obstacle surfaces are approximated by grid cells like a step function. Following the 2.5 D approach, the model domain can be categorized in three subdomains:

(i) grid cells in free atmosphere without adjacent solid surfaces,
(ii) grid cells within orography,
(iii) grid cells adjacent to solid surfaces , where a local surface layer is assumed.

A sketch of the topography implementation is presented Figure 2. For fluid cells, the normal PALM code is executed. In the region between inner and outer surface boundaries, wall functions are applied to the surface layer. We used Neumann no-slip condition combined with Monin-Obukhov similarity theory (MOST) that assumes a constant flux layer between inner surface boundary $z(k = 0)$ and $z(k = 0) + 0.5\Delta z$ (the first computing grid point at cell centre for horizontal wind components and scalars). When considering the yellow solid cells, the PALM standard code is executed but multiplied by zero. Note that the surface on solid cells on the horizontal plane is defined by the location on wall-normal velocity, respectively on grid cell surfaces. Furthermore, the grid cells surrounded by three or more walls are filled by the lowest height of neighbouring grid cells. This filtering function aims to avoid the case that a single surface layer cell is surrounded by three or more solid cells, where numerical instabilities could occur.

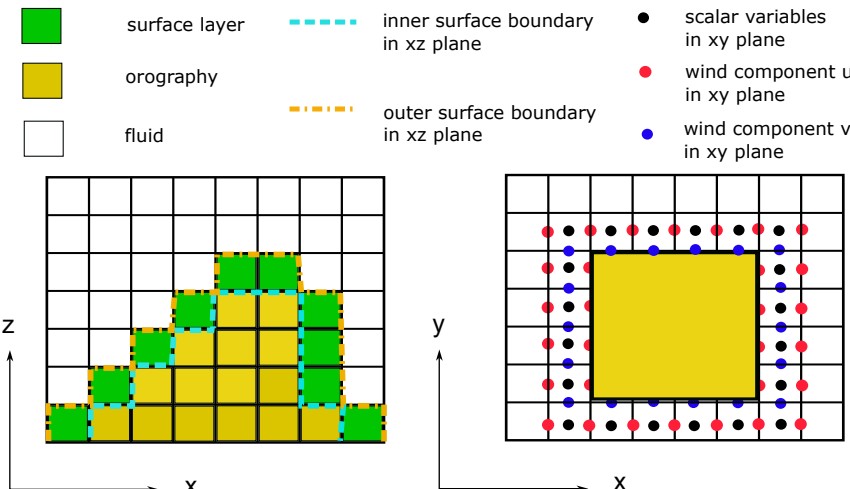

**Figure 2.** Sketch of the Cartesian topography in PALM. The yellow cells represent orography, the white cells indicate fluid cells and the surface layer adjacent to orography are displayed in green on xz plane. The light blue lines represent the inner surface boundary of the surface layer on xz plane, while the orange lines defines the outer boundary on xz plane . According to the definition of the staggered C-grid, the black points on xy plane represent cell centres, where scalar variables are masked. The locations of *u* and *v* wind components displayed in red and blue define the location of impermeable orography surface.

To define the topography information, PALM requires a static driver file in a rastered format such as Network Common Data Form (NetCDF). The topography implementation in PALM conforms with the Digital Elevation Model (DEM) format, which is offen used in geographic information systems to represent terrain. We used the Shuttle Radar Topography Mission (SRTM) elevation data, which were downloaded via the open-source software Quantum Geographic Information System (QGIS), originally in a resolution of 30-m. The geodata were further downscaled in 5 m to resolve small-scale turbulences in a range similar to the wingspan of the considered airplane Piper PA-28. A pre-processing tool named palmpy for generating static driver file was implemented in the frame work of a master thesis in the Centre for Aviation (ZAV) at the Zurich University of Applied Sciences [66]. The in-house script is comparable with the original routine included in PALM model system, but was adapted specifically for the aeronautical applications in the ZAV.

3.4.2. Mesoscale Nesting and Synthetic Turbulence Generation

PALM version 6.0 was extended to simulate atmospheric boundary layers under evolving synoptic conditions. A preprocessor INIFOR (initialization and forcing) included in PALM model system provides initial and time-dependent lateral and top boundaries

based on the regional weather model COSMO. Further, additional syntethical turbulence fluctuations are imposed on the lateral boundaries. At the top boundary, Dirchlet free-slip conditions are used for velocities, and scalars are defined under Neumann conditions with temporal constant gradients and zero-gradients in *z* direction. In this section, we introduce the mesoscale nesting interface and the implemented synthetic turbulence generator briefly. A comprehensive description of this interface including verification study were published in [46].

The non-hydrostatic model COSMO is used operationally all around the globe, however with a strong focus on Germany and Switzerland. There, the COSMO model is run in different resolutions, where the highest resolution currently used Switzerland is of 1.1 km (COSMO-1: 1.11 km N-S and 0.75 km E-W) and spreads over 80 vertical coordinate levels in a smooth level vertical (SLEVE) terrain-following coordinate [67]. The COSMO-1 reanalysis (historic) data is available in hourly resolution on-demand and was ordered from the Federal Office of Meteorology and Climatology in Switzerland (MeteoSwiss) for the present study.

Figure 3 presents the workflow of using the mesoscale nesting interface INIFOR for a PALM simulation. INIFOR has been developed mainly by the German Meteorological Service (DWD) and is currently designed to process the output data of DWD's operational configuration COSMO-D2 and its predecessor COSMO-DE [46]. Due to the slight difference between COSMO-1 and COSMOS-D2/DE data in coordinate system, COSMO-1 reanalysis data is to be converted in a conform form prior to importing into the nesting interface. INIFOR reads COSMO data and a static drive file containing topography information, and forces interpolated COSMO data in a rotated-pole system onto PALM's Cartesian grid. INIFOR generates a dynamic driver file as output. The dynamic driver file is required by PALM as simulation input, when enabling the mode mesoscale nesting, so called "offline-nesting" in PALM model system.

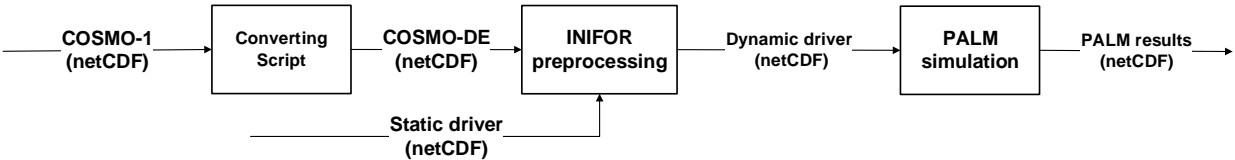

**Figure 3.** Simulation workflow of mesoscale nesting.

In order to accelerate the transition of the interpolated mesoscale wind fields on the lateral boundaries to a fully developed turbulence atmospheric boundary layer, PALM allows to generate synthetic turbulence on the inflow boundary. Based on a Cholesky decomposition suggested by Lund et al. [36], the turbulent flow velocity $u_i$ on the 2D boundary plane consists of the mean inflow value $\overline{u}_i$ and the fluctuations

$$u_i = \overline{u}_i + a_{ij}u_j'', \tag{69}$$

with the amplitude tensor $a_{ij}$ defined as

$$a = \begin{pmatrix} \sqrt{r_{11}} & 0 & 0 \\ r_{21}/a_{11} & \sqrt{r_{22} - a_{21}^2} & 0 \\ r_{31}/a_{11} & (r_{32} - a_{21}a_{31})/a_{22} & \sqrt{r_{33} - a_{31}^2 - a_{32}^2} \end{pmatrix}. \tag{70}$$

Here, $i, j \in \{1, 2, 3\}$. $r_{ij}$ is the Reynolds-stress tensor. $u_j''$ are the spatial and temporal disturbances imposed onto the 2D plane. Xie and Castro [40] suggested a 2D random

signal $\Psi_i$ correlated with length scales $L_i$ and timescales $T_i$ on the 2D plane in each velocity component. The disturbance term $u_i''$ computes as

$$u_i''(t + \Delta t) = \Psi_i^{t-\Delta t}(L_i) \exp\left(-\frac{\pi \Delta t}{2T_i}\right) + \Psi_i^t(L_i)\left[1 - \exp\left(-\frac{\pi \Delta t}{T_i}\right)\right]^{0.5}, \tag{71}$$

with $\Delta t$ being the simulation timestep.

In a later paper [41], Kim et al. performed a correction to the velocity $u_i$ in order to maintain constant mass flux normal to the considered 2D Plane. In addition, to avoid the perturbations having non-zero mean $u_j$ and $u_k$ components on the west model boundary [46], the disturbance term $u_i''$ is corrected as follows

$$u_{i,\text{corr}}'' = u_i'' - \frac{1}{S}\int_{\partial S} u_i'' \mathrm{d}S \tag{72}$$

with $S$ being the surface area of the 2D boundary.

The remaining unknown Reynolds-stress components $r_{ij}$ and scale length $L_i$ can be either pre-defined in a ASCII simulation input file or parameterized. For a detailed description of the parameterization, we refer to [46].

*3.5. Grid Nesting and Computational Domain*

To reduce computing costs of LES simulation involving a large computational domain, PALM provides an online LES-LES nesting system. It allows multiple child domains with high grid resolution nested in a parent domain of a PALM simulation. The nesting approach can be further categorized in two modes: (i) one-way coupling; (ii) two-way coupling. In the first case, the information derived from parent domains is interpolated on grid points of child domain(s). The second approach uses an anterpolation algorithm to transfer the data from child domain back to the corresponding parent domain. A detailed description was presented in [68]. We used the cost-effective one-way coupling, since the modest systematic shift in streamwise velocity indicated in [68] can be acceptable for our application.

Figure 4 shows the computational domain, which locates mainly in the region Samedan in Switzerland, including a small part within Italy on the south side of Piz Bernina (4049 m). The case discussed in the present article contains a parent domain with the domain origin at 9.95° E, 46.33° N in the global coordinate system WGS84, 774,572 E, 133,915 N in the Swiss cooridnate system $CH1903+$, and extends over 24,576 × 133,915 × 4096 m$^3$ in the $x$, $y$ and $z$ direction, with an equidistant grid spacing of 32 m. The x direction follows E-W axis and the y direction is parallel to S-N axis in the global coordinate system. In the vertical z direction, the bottom of the parent domain locates at 1746 m AMSL, which is determined by the topography information included in the dynamic driver. Further, two child domains of 2048 m × 2048 m × 3584 m were nested in the parent domain, The both child domains start from the parent domain bottom in the z direction and have a grid resolution of 8 m. We simulated the weather conditions of a summer day from UTC 06:00–12:00. Table 2 lists the information of the computational domain configuration.

**Table 2.** Computational domain configuration.

| Spatial Domain | Number of Grid $(n_x, n_y, n_z)$ [−] | Grid Resolution $(d_x, d_y, d_z)$ [m] | Domain Origin in Coordinates System $CH1903+$ | $WGS84$ | Time Domain |
|---|---|---|---|---|---|
| Parent domain | (768, 576, 128) | (32, 32, 32) | (774,572 E, 133,915 N) | (9.71° E, 46.33° N) | |
| Child domain 1 | (256, 256, 448) | (8, 8, 8) | (778,864 E, 140,087 N) | (9.76° E, 46.39° N) | UTC 06:00–UTC 12:00 |
| Child domain 2 | (256, 256, 448) | (8, 8, 8) | (793,612 E, 141,947 N) | (9.96° E, 46.40° N) | |

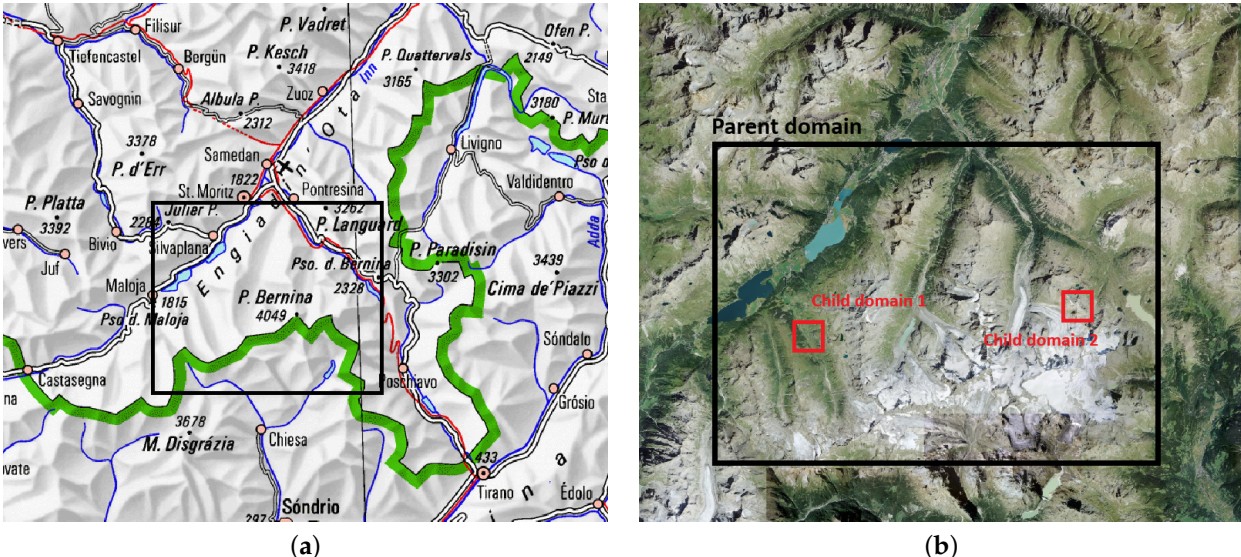

**Figure 4.** PALM simulation domain for the mountainous region Samedan indicated in (**a**) national map of Switzerland with relief, in (**b**) SWISSIMAGE (source of the original images: the Federal Office of Topography in Switzerland). The border between Switzerland and Italy is displayed in green in (**a**). The black frame in each subfigure represents the parent domain defined in the present case. The red frames in (**b**) indicate the child domains of the parent domain.

## 4. Flight Simulation Test Campaign

A flight simulation environment was created in the ZAV's Research and Didactics Simulator (ReDSim) to evaluate the first implementation of the energy management system [1]. The ReDSim is a real-time didactics and research simulator that was developed at the ZHAW. The research simulator employs state of the art technology and features a Control Loading System (CLS) to simulate force feedback to the pilot and a visual system with a field of view of 180 degrees. ReDSim is utilized in research projects and taught courses at the ZHAW as laboratory equipment, where students can gain hands-on experience with flight simulation devices for research. Additionally, ReDSim is utilized for research and development projects of external industry partners.

To evaluate the concept of the developed energy management system, we ran a series of flight simulation experiments at the beginning of 2021. A group of experienced pilots was invited to test the new system's initial installation. The pilots tested the functionality of a graphic user interface (GUI) included in the energy management system and provided feedback on its user-friendliness using a standardized questionnaire [1]. Our most recent flight simulation experiments focused on the integrated atmospheric disturbance model. The test session described in this article was conducted by three private pilots and an experienced meteorologist who also holds a private pilot license (PPL).

### 4.1. Concept of Flight Simulation Test Environment

Figure 5 shows the overview of the flight simulation test environment. The flight simulation model is based on the ReDSim existing PA-28 real-time simulation model. Several adjustments were made, however, to strengthen the model and fit the simulation environment for the concept study. The new subsystem outputs depicting atmospheric disturbance were included into the environment model and aircraft equations of motion (EoM). The atmospheric turbulence disturbance can be generated in two ways:

(i)    by importing PALM instantaneous data with high temporal resolution with local mesh refinements (hereafter called turbulence case PALM),

(ii)   by activating Dryden turbulence model plus temporally mean data from PALM simulations without local mesh refinements (hereafter called turbulence case Dryden).

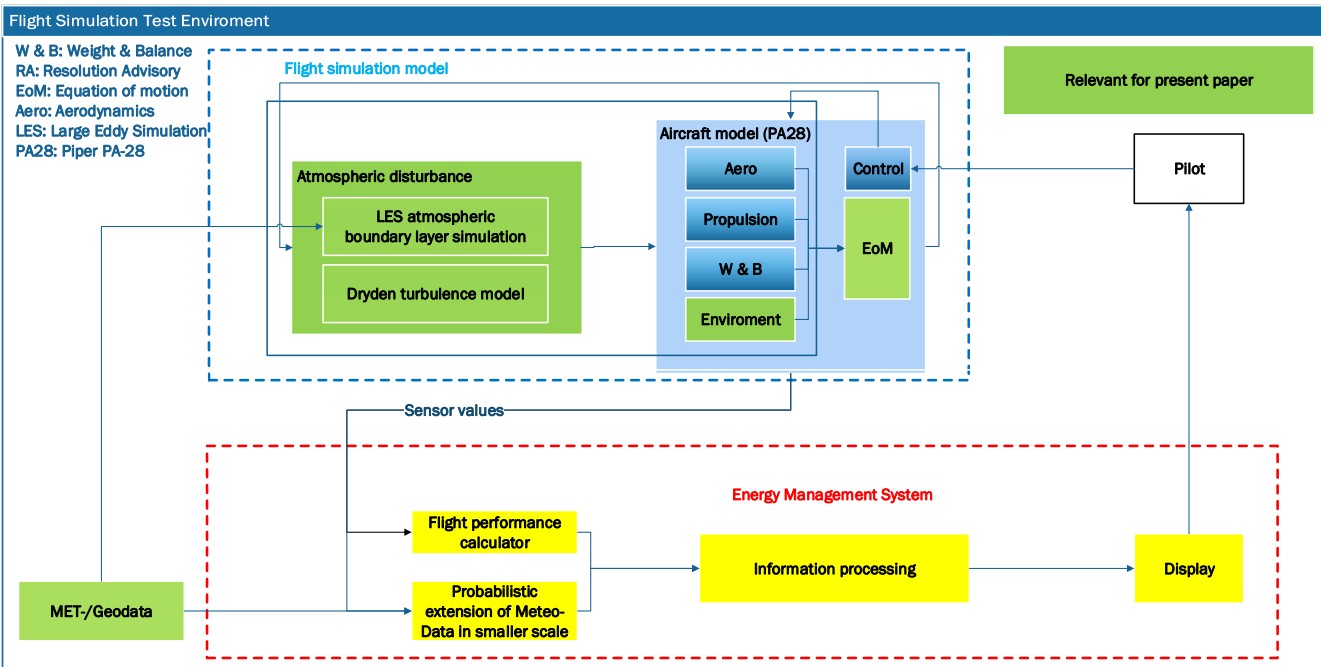

**Figure 5.** Overview of the piloted flight simulation environment. The green blocks represent modules associated with the model of atmospheric disturbance, which is a submodel of the flight simulation model. The yellow blocks represent the energy management system's subsystems.

Prior to starting the flight simulation, the PALM output data is imported into MATLAB (in netCDF format). The wind velocity components $u, v, w$, atmospheric pressure $p$ and temperature $T$ (converted from the original PALM output potential temperature $\theta$) are interpolated as linear functions of the space coordinate $x, y, z$ and of time $t$. The output of this procedure is saved as gridded interpolants in the MATLAB workspace, which the flight simulation model can later access. Wind velocity components, pressure, and temperature are estimated as functions of aircraft position and time throughout the real-time simulation. Additionally, the wind angular rates $p, q, r$ at any instant are calculated as

$$p = \frac{\partial w}{\partial y} - \frac{\partial v}{\partial z} \tag{73}$$

$$q = -\frac{\partial w}{\partial x} + \frac{\partial u}{\partial z} \tag{74}$$

$$r = \frac{\partial v}{\partial x} - \frac{\partial u}{\partial y} \tag{75}$$

where the derivation is performed numerically through centered differences.

### 4.2. Summary of Test Procedure

During the previous test session [1], we conducted the flight simulation experiment in accordance with the industry standard procedure for airplane certification. The experiment was divided into two parts: (i) testing of engine performance and flight performance test in calm air condition, and (ii) testing of simulation environment with enabled atmospheric disturbance subsystem.

The present test session can be considered as an extension experiment of the second part in the last session, with an emphasis on turbulence structure and its effect on airplane response in mountainous region. Furthermore, the test campaign aims to investigate and compare the two approaches to generating turbulent velocities in the flight simulator.

Figure 6 shows two scenarios discussed in the present study. Prior to conducting the piloted flight simulation test, we performed offline simulation test along a valley lying west of the weather station Piz Corvatsch (3294 m), which is a satisfactory location for undisturbed free atmosphere benchmaking . We used an in-house developed autopilot to perform several three-minute cruise climb manoeuvres, with heading 155° from 7000 ft (2134 m) to 8000 ft (2438 m). For the piloted flight simulation test, the pilots performed two to three runs, each of ten to fifteen minutes, utilizing two different methods for generating atmospheric disturbances. Each run required the pilot to complete a tracking task. After trimming the aircraft, the tracked flight began over the lake Bianco. The pilots made several level turns before climbing steadily into the child domain 2 over the glacier at around 11,000 ft (3350 m). Subsequently, further level turns were performed over the glacier. After each run, the pilots were asked for their comments on turbulence by means of a questionnaire. We took the question sheet included in [7] as reference and adapted it for our experiment in a fixed base flight simulator. This question sheet asked the pilots to evaluate the turbulence characteristics, such as turbulence intensities and patchiness of the turbulence, airplane responses due to the disturbances, and workload required for performing the manoeuvres level turn and steady climb by using Cooper rating scale [69]. The relevant documents, weight and balance sheet and flight test card and the questionnaire are presented in the Appendix A.

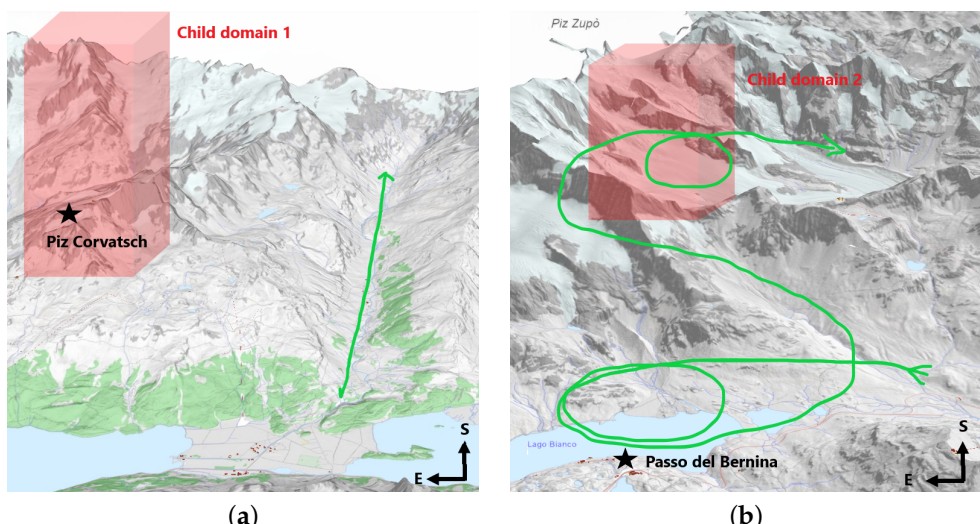

**(a)**                    **(b)**

**Figure 6.** Test scenarios marked in the Swiss federal 3D map viewer for (**a**) the offline flight simulation and (**b**) the piloted flight simulation (source of the original images: the Federal Office of Topography in Switzerland). The green curve in each subfigure represents the hypothetical flight path. The red shadowed zones represent the fluid cells of the child domains of the PALM simulation, which are defined in Figure 4 and Table 2. The black stars indicate the weather stations Piz Corvatsch 3294 m and Passo del Bernina 2260 m Bernina.

## 5. Results and Discussion

In the following section, we show the results of the PALM simulations and the flight simulation experiments. As a guide for the readers as they proceed through this section, we outline the section's structure here. As a starting point, we evaluate the PALM simulations in terms of revealing the structure of the convective boundary layer over the whole calculation domain, presenting temporal evolution of mean wind velocities at the areas of interest in Section 5.1. In the following Section 5.2, we describe the results obtained from flight simulation experiments, with focus on the differences between the two ways to generate atmospheric disturbance in frequency domain. Finally, we provide the results of the piloted flight simulations for the chosen scenario and analyze pilot feedback on the atmospheric disturbance model.

### 5.1. PALM Simulation Evaluation

5.1.1. Vertical Profiles

To give an overview of the atmospheric boundary layer (ABL) simulated by PALM, we present in this section the hourly averaged vertical profiles of various horizontal mean quantities over the whole parent domain.

Figure 7 shows the horizontal mean values of horizontal wind speed $u_h$, vertical wind speed $w$ and potential temperature $\theta$ starting at 06:00 UTC on 4 August 2017, which was a warm summer day characterized by clear-sky conditions. The horizontal wind speed profiles show a characteristic velocity profile within the convective boundary layer for the six hours studied. The mean value between 2000 m and 3500 m, where flight simulation tests were performed, is less than 5 ms$^{-1}$. The low range of vertical speed indicates that the vertical motions of air parcels within the boundary layer are relatively slow. The stably stratified layer is confirmed by the vertical temperature profile as well. During the temporal evolution, due to the heating of the surface, the boundary layer continuously warms up and the potential temperature profiles shift to the right with a quasi-constant temperature gradient in vertical direction. In the first three hours (UTC 06:00–UTC 09:00), the negative temperature gradients indicate an unstable stratification near the surface, where turbulent wind can be induced.

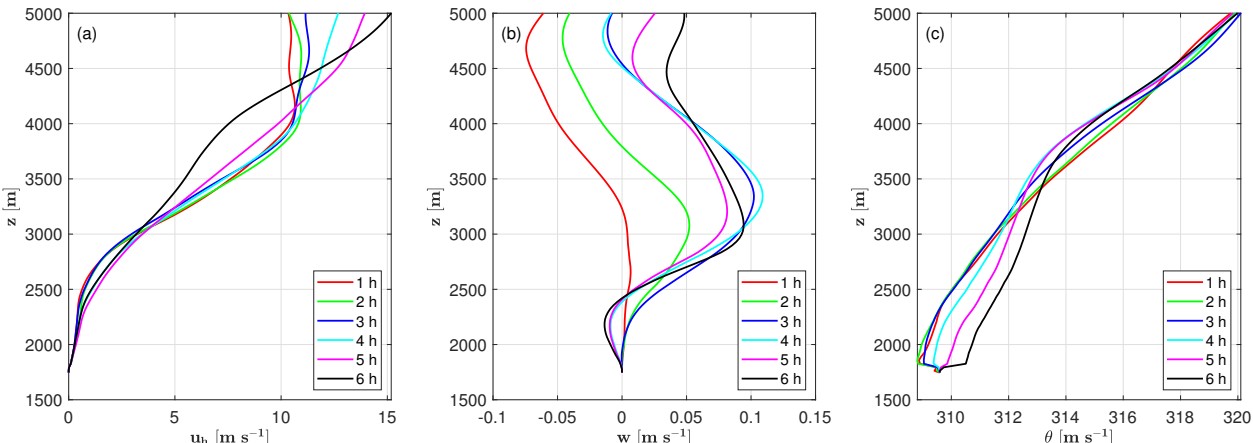

**Figure 7.** Horizontal mean vertical profiles of (**a**) horizontal wind speed, (**b**) vertical wind speed and (**c**) potential temperature at the whole domain in time after the start of the simulation at 06:00 UTC.

Figure 8 shows the horizontal mean vertical profiles of heat-fluxes. It can be seen that the flow is well resolved throughout the boundary layer, while sub-grid scale fluxes dominate in the near-surface layer. The vertical profiles of total heat-fluxes indicate the boundary layer height $\Delta z$ of about 2000 m. The different signs of the vertical gradients of the heat fluxes gradient can be explained by heating the lower part of an inversion layer and cooling of the upper part.

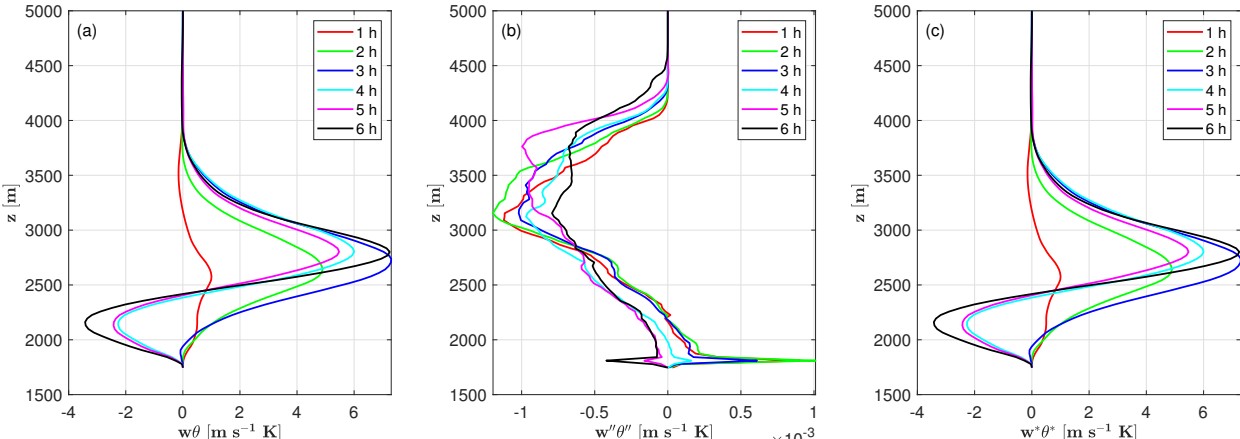

**Figure 8.** Horizontal mean vertical profiles of (**a**) total heat flux, (**b**) sub-grid scale (SGS) heat flux and (**c**) resolved-scale heat flux at the whole domain in time after the start of the simulation at 06:00 UTC.

Figure 9 shows the vertical profiles of horizontally averaged flux components $wu$, $wv$ and subgrid-scale turbulent kinematic flow $e$ over the whole domain at UTC 07:00, UTC 09:00 and UTC 12:00. Following about one-hour spinup time, the differences in fluxes between the two SGS models are determined to be insignificant, while SGS turbulent kinematic energy values calculated using the Deardorff SGS model are clearly higher than those calculated using the dynamic SGS model, in particular near the wall. This impact can be described as the Deardorff model not being capable of resolving as many small eddies near the wall as the dynamical SGS model, when same grid spacing is used.

The vertical profiles of variances horizontally averaged over the whole domain are plotted in Figure 10. During the time revolution, the maximal values remain at the same altitude. Similarly, there are no significant differences between the two models in computing velocity variance, once the simulation is stabilized.

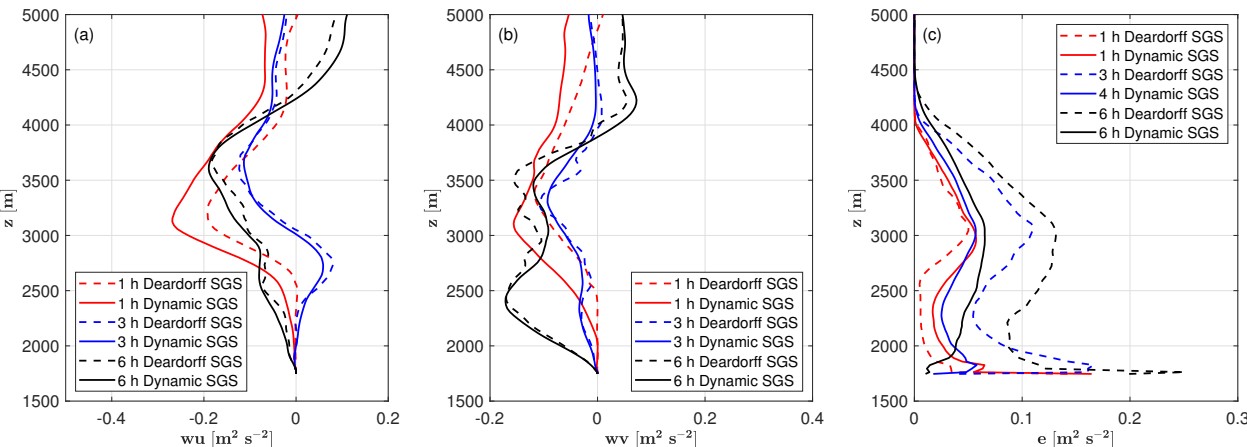

**Figure 9.** Horizontal mean vertical profiles of (**a**) flux component $wu$, (**b**) flux component $v$ and (**c**) subgrid-scale (SGS) turbulent kinetic energy $e$ at the whole domain in time after the start of the simulation at 06:00 UTC. The dashed curves indicate the simulated values by using Deardorff SGS model. The continuous curves present the simulated values by using the dynamic SGS model.

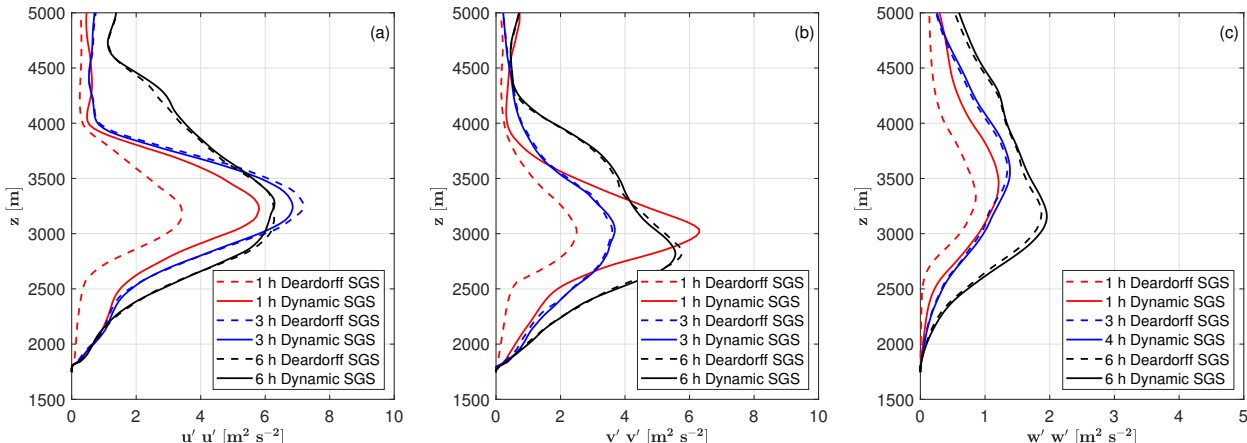

**Figure 10.** Horizontal mean variances of (**a**) wind component *u*, (**b**) wind component *v* and (**c**) wind component *w* at the whole domain in time after the start of the simulation at 06:00 UTC. The dashed curves indicate the simulated values by using Deardorff SGS model. The continuous curves present the simulated values by using the dynamic SGS model.

### 5.1.2. Temporal Evolution

In this section, we show the time evolution of the wind fields within the boundary layer in this section. To evaluate the LES simulations' performance, measured data on horizontal wind components were used. We collected data from the SwissMetNet (SMN) network of weather stations. The Swiss Federal Office of Meteorology and Climatology maintains the national meteorological and climatological network that includes over 150 stations recording wind direction and speed. The measurements are utilized for a variety of purposes, including monitoring basic meteorological parameters, issuing meteorological warnings, providing input for numerical weather forecasting, and conducting climate analysis. For a detailed description of the network, we refer to [70].

Several SwissMetNet stations were accessible for benchmarking purposes in the selected area of study, However, only one station (COV/Piz Corvatsch, (774,572 E, 133,915 N) was located on a mountain ridge at an altitude of 3294 m (783,145 E, 143,524 N), as a satisfactory station representing best the state of the undisturbed free atmosphere (shown in Figure 6a). The data of COV was received through an interface named IDAWEB, which provides free access to surface meteorological measurement data from the SwissMetNet and other networks to research organizations. In addition to the SwissMetNet data, the measurements at the station Passo del Bernina (BEH, 798,422 E, 143,020 N) at 2260 m were taken from Intercantonal Measurement and Information System (IMIS) operated by Swiss Federal Institute for Forest, Snow and Landscape Research (WSL). To compare with the experimental data, we defined multiple wind measurement masts near the considered weather stations. The simulated data shown in the following are spatially averaged values over a virtual measuring box of approximately 100 m × 100 m × 40 m.

Figure 11 shows temporal evolutions of 10 min time-averaged horizontal wind speed from the measurements and the PALM simulation at the weather stations Piz Corvatsch and Passo del Bernina, over the total simulation time period of 21,600 s. Both series of simulated horizontal wind speeds (Deardorff SGS model and dynamic SGS model) in black present temporal evolutions that are consistent with those of the measured values at both weather stations. The root mean square error over the whole simulation time is approximately 2 m s$^{-1}$, which is acceptable, even if the measured data indicate a greater bias than the simulated data. Furthermore, in the one-hour time frame of interest for the flight simulation denoted in light blue shading, the root mean square error is less than 1 m/s. These comparisons demonstrate the LES simulation validity for estimating mean wind fields in areas like COV with an undisturbed free atmosphere as well as in orographically complex regions such as BEH. In Figure 11, it can be noticed that there is no

significant difference in time domain between the two SGS models when estimating mean wind fields. Hence, we use dynamic SGS model as default simulated data in the following.

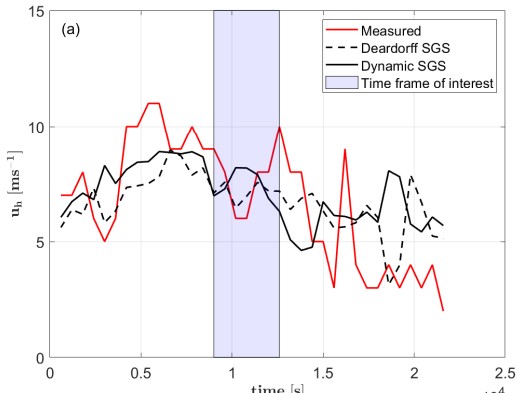 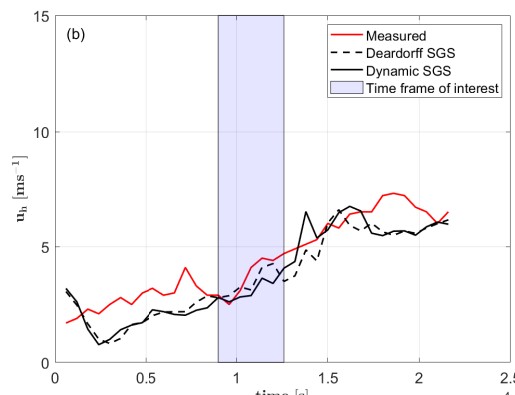

**Figure 11.** Temporal evolutions of 10 min time-averaged horizontal wind speed $u_h$ at (**a**) weather station Piz Corvatsch (COV) and (**b**) weather station Passo del Bernina BEH over 21,600 s simulation time. The red lines present the measured data. The simulated data of Dynamic SGS model and Deardorff model (dashed) are displayed in black. The box in both subfigures in light blue shading indicate the time frame of interest for the flight simulation experiment, from 9000 s to 12,600 s after the start of the simulation at 06:00 UTC.

The temporal evolutions of wind direction at both weather stations are shown in Figure 12. At the weather station COV, the simulated wind direction time histories, match well the available hourly time-averaged data, denoted by the red circle. On the test day, the wind fields in this location were relatively stable. At the weather station BEH, the recorded wind direction changes relatively little during the observed six hours, whereas the simulated data indicate tremendous oscillations across the whole time domain. This can be explained as a complex boundary layer structure containing various small eddies. Despite this, the hourly averaged wind direction of 152° from the simulation indicates a good agreement with the measured averaged value of 141° in the selected hour.

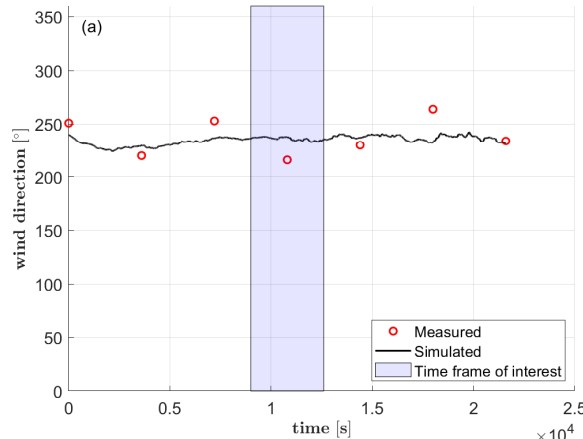 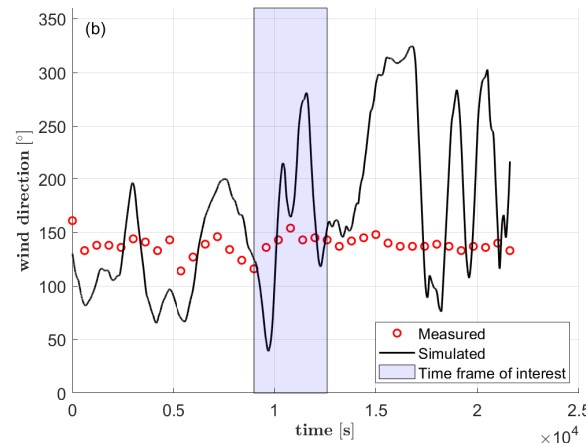

**Figure 12.** Temporal evolutions of wind direction at (**a**) weather station Piz Corvatsch (COV) and (**b**) weather station Passo del Bernina (BEH) over 21,600 s simulation time. The red scattered points solid lines present the hourly time-averaged measured data at COV and the 10-min averaged data at BEH. The solid lines represented 10 min time-averaged wind direction derived from the simulation. The box in both subfigures in light blue shading indicate the time frame of interest for the flight simulation experiment, from 9000 s to 12,600 s after the start of the simulation at 06:00 UTC.

In order to give an overview of the spatial structure of the wind fields, we plot in Figure 13 the instantaneous vertical wind speed over the 3D calculation domain on the top of a landsat satellite image at UTC 09:00, 10,800 s after the start of the simulation. The weather stations are marked by green (COV) and yellow (BEH) stars. The two solid lines indicate the positions in x direction of the y-z cross-sections plots shown in Figure 14. In the region near Piz Corvatsch, the 3D plot shows a regular wave structure near the weather station COV, which is mainly oriented along the mean-wind direction of 225° (see Figure 12) with "up- and downdrafts" on the order of 2 m s$^{-1}$. The wavelength is estimated visually to be $\mathcal{O}(\sim 5$ km$)$. The wavelength near the weather station BEH is clearly shorter.

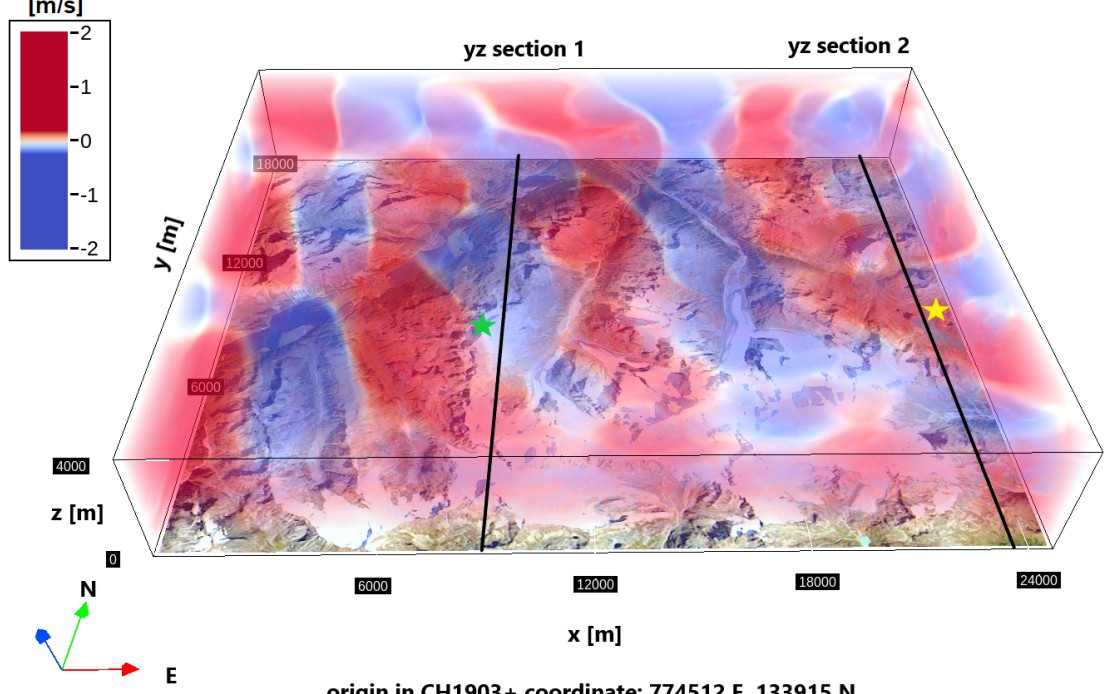

**Figure 13.** Instantaneous vertical wind speed over the whole 3D domain with the original located at 10,800 s after start of the simulation at 06:00 UTC. The 3D volume rendering is displayed over the landsat satellite image of the computation domain (source of the original images: the Federal Office of Topography in Switzerland). The solid black lines denote the locations the y-z cross-section plots shown in Figure 14, at x = 8704 m (position 1) and x = 23,840 m (position 2) in the local simulation coordinate.

Figure 14 shows the Instantaneous y-z cross-sections of vertical wind speed and potential temperature at 09:00 UTC at positions x = 8704 m and x = 23,840 m. The approximately horizontal isolines of potential temperature represent the stable boundary layer structure in the 8000 m to 10,000 m y direction, near the location of COV, while the chaos flow structure extend to the altitude 4500 m, in the same range in y direction, where weather station BEN locates. Additionally, Figure 15 provides a temporal evolution of potential temperature and horizontal flow fields on x-y sections between UTC 09:00 to UTC 09:30 at the altitude 3200 m, in the 8000 m to 10,000 m y direction. As illustrated in the graphs, the flow fields and temperature distribution vary often in this local region. This represents graphically the severe oscillations of wind direction at the weather station BEH, indicated in Figure 13. These complex flow structures could be realistic or just pure numerical outcomes. In the following sections, we will continue the discussion about the realism of the atmospheric turbulence in frequency domain, by examining the flight simulation outputs.

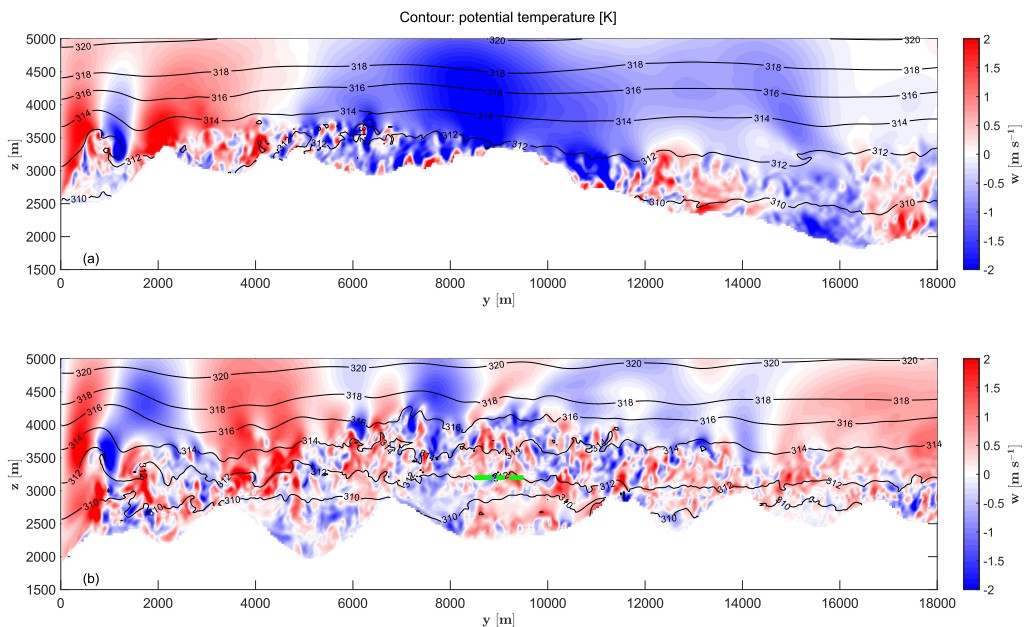

**Figure 14.** Instantaneous y-z cross-sections of vertical wind speed $w$ and potential temperature $\theta$ (contours) at 10,800 s after the start of the simulation at 06:00 UTC, taken at (**a**) x = 8704 m (position 1) and (**b**) x = 23,840 m (position 2) in the local simulation coordinate. The green dashed line in (**b**) indicated the position of x-y cross sections shown in Figure 15.

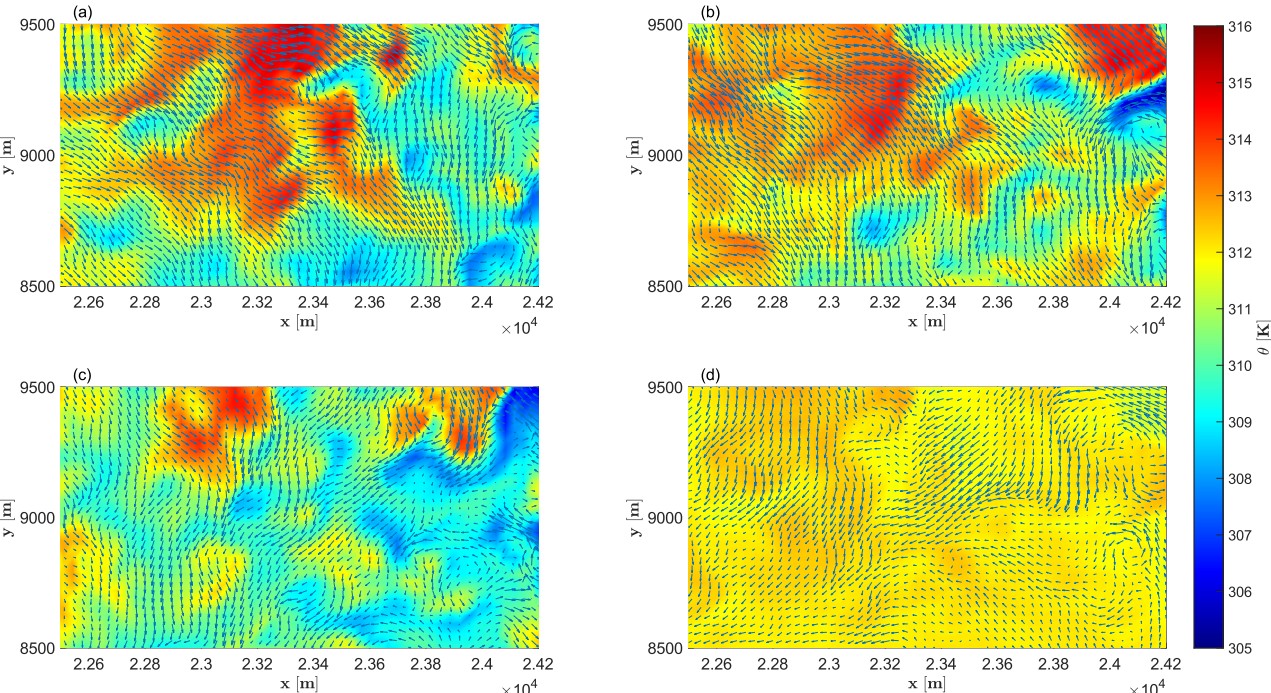

**Figure 15.** Instantaneous x-y cross-sections of potential temperature and horizontal flow fields (vector plots) at (**a**) UTC 09:00, (**b**) UTC 09:10, (**c**) UTC 09:20 and (**d**) UTC 09:30 at the altitude of 3200 m.

## 5.2. Results of Flight Simulation Test

A series of flight simulation tests was conducted with the purpose of assessing the realism of two approaches to simulate atmospheric disturbance. The first subsection compares the power spectral densities of wind velocity and airplane angular rate responses

in the two scenarios: (i) Piz Corvasch with an undisturbed free atmosphere, and (ii) Passo del Bernina with complex terrain (see Figure 6). The second subsection contains the results of the pilot questionnaire.

5.2.1. Spectral Analysis

Many research investigations on the atmospheric boundary layer (ABL) and turbulence spectral forms have been conducted in the domains of geophysics and atmospheric sciences. Some major hypotheses were used in combination with our specific test cases to narrow the scope of the discussion about the extensive topic and essentially establish a realistic and sufficient strategy to generate atmospheric disturbance for our research work related to flight safety in mountainous terrain.

Normally, the motions of turbulent eddies are described as an energy cascade from larger to smaller scales through the interactions of eddies. Depending on the scales, the eddies can be categorized in three subranges: energy containing scale (large scale), initial scale and dissipation scale (small scale). The later two subranges are referred to equilibrium range. The horizontal axis of the energy spectrum of turbulence denotes eddy frequency $f$ or wave number $k = 2\pi/L$ in $m^{-1}$, where $L$ is the length scale. In our case, the variable $L$ can be calculated as $L = f/V_{tas}$, where $V_{tas}$ in $m\ s^{-1}$ is the true airspeed the airplane. According to the Kolmogorov's isotropic turbulence hypothesis [71], the velocity spectral of isotropic turbulence follows a $-5/3$ law scaling the inertial subrange. Lindborg [72] found that the $-5/3$-law could be applied for anisotropic turbulence, e.g., stably stratified ABL. The lower side of anisptropic inertial range in the energy spectrum was characterized by Dougherty-Ozmidov eddy scale $k_o$ [73,74]. The opinion in this paper was supported by various studies of direct numerical simulations. In a recent study, Cheng et al [75] presented a comprehensive discussion about the spectral shape in the stable ABL and proposed a new hypothesis for stable ABL. The inertial subrange in the equilibrium range could be separated by a larger scale $k_b$ (buoyancy wave number) and smaller scale $k_o$ (or an antitropic scale $k_a$) into three regions: buoyancy subrange, a transition region and the isotropic inertial subrange. In first and the third region, the energy spectrum followed the $-5/3$-law, while a shallower $-1$-slope was presented in the transition region. Our present study doesn't cover the entire range of eddy scale, but rather focuses on a relevant range for flight safety study. Thus, based on the founding from Knigge et al [76], we assumed that length scale between 5 m and 50 m might have dominant impacts on a general aviation airplane with about 10 m wingspan.

Figures 16–19 present the results of the spectral analysis. Besides the $-5/3$-slope according to Kolmogorov's law [71], either a $-1$-slope or $-2$-slope is plotted as a reference line in each logarithmic chart. In order to better identify the spectra slope and compare with references, the vertical axis in the following plots denotes the product of wave number $k$ and power spectral density $\Phi$.

Figure 16 shows the spectra of wind velocity components $u$, $v$ and $w$ during a steady climb flight at Piz Corvatsch with a grid resolution of 32 m in the PALM simulation. The turbulence case Dryden presents isotropic turbulence, whereas the vertical wind velocity energy spectrum in the turbulence scenario PALM is clearly lower than the horizontal spectra. It can be observed that the spectra in case Dryden are higher than the theoretical values following the $-5/3$-law in the considered wave number range. Spectra of case PALM show a different shape; the energy is transported dramatically from large scale eddies to the small scale ones, which is considered not realistic. This is a typical result derived from LES simulation, in which the cut-off length for SGS-model is larger than the higher limit of the wave number $k_{max}$. The underestimation of the energy for small eddies is caused by the numerical errors in SGS model.

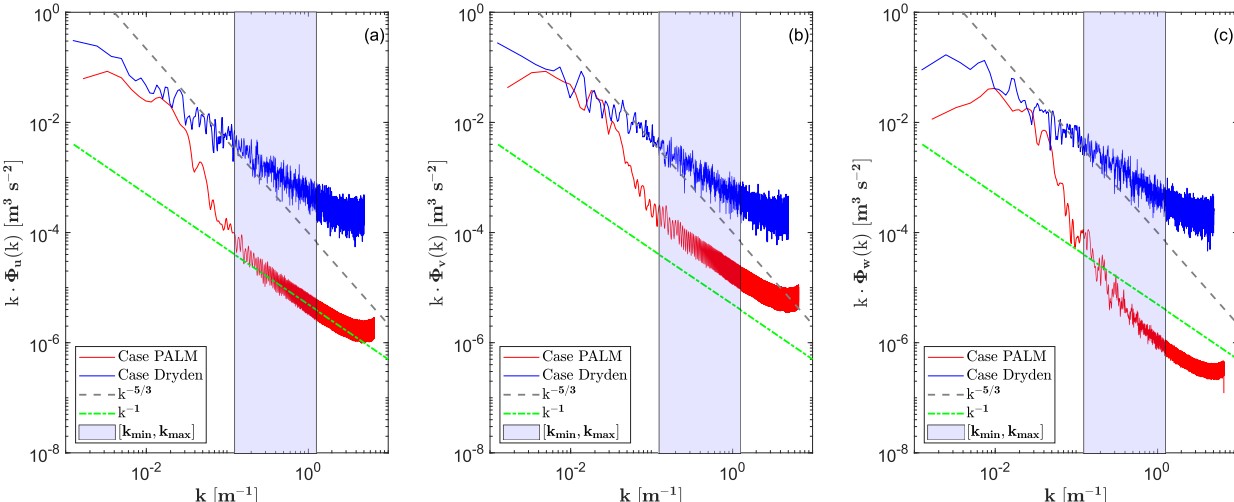

**Figure 16.** Spectra of wind velocity component (**a**) *u*, (**b**) *v* and (**c**) *w*, derived from flight simulation output data of scenario Piz Corvatsch at an average altitude of 7500 ft (2280 m). The grey dashed line represents the reference $-3/5$-slope according to Kolmogorov's law [71]. The $-2$-slope is plotted in green. The box in light blue shading indicates the time frame of interest for the flight simulation experiment, which is determined by the lower limit wave number $k_{min}$ of $2\pi/50$ m$^{-1}$ and the high limit wave number $k_{max}$ of $2\pi/5$ m$^{-1}$.

In Figure 17, we compare both turbulence cases for predicting spectra of airplane angler rate *p* and *q* during the same flight in the region Piz Corvatsch. In general, the simulated angular rates under both Dryden and PALM turbulence conditions show steeper slopes than the theoretical value of $-5/3$. The deficiencies might be linked to the airplane model itself or other components, which will not be further discussed. The analysis will be focused mainly on the difference between two turbulence cases. It can be seen the both cases have a good agreement in the spectrum of the roll rate *p*, while spectra of *q* and *r* derived from turbulence case Dryden are higher than the ones of turbulence case PALM. The lower values can be explained by the overestimated dissipation rate of SGS eddies in the PALM simulations.

In Figures 18 and 19, we present the spectra calculated by using the flight simulation test data of scenario Passo del Bernina. Figure 18 shows the spectra of wind velocity components *u*, *v* and *w* recorded during the level turns with various bank angles above the glacier (Figure 6b), mostly within the child domain 2 with a grid resolution of 8 m in the PALM simulation. In comparison to the Piz Corvatsch scenario at 7500 ft, an isotropic spectrum with a comparable shape and value can be detected at the altitude of 10,500 ft in case Dryden. The spectra in case PALM show reasonable shapes and can follow the theoretical $-3/5$-slope. The prediction in the small scale region was improved by using the fine mesh. Furthermore, the shallower slope of about $-1$ could be explained by the hypothesis proposed in [75], although the effect of the transition region is not as obviously as shown in their study on a strongly stratified ABL. The drops at about $k = 1$ and oscillations in the small-scale range might be the result of multiple level rotations with a large radius and a small bank angle that fly outside the child domain. The corresponding spectra of angular rates are plotted in Figure 19. It can be seen the spectra of roll rate *p* and yaw rate *r* are consistent in both cases. The spectra of pitch rate *q* in case PALM lies still under the one derived from case Dryden, but the dissipation rate is lower than in the scenario Piz Corvatsch. This improvement in the small scale region can be explained by the finer spatial resolution in the child domain.

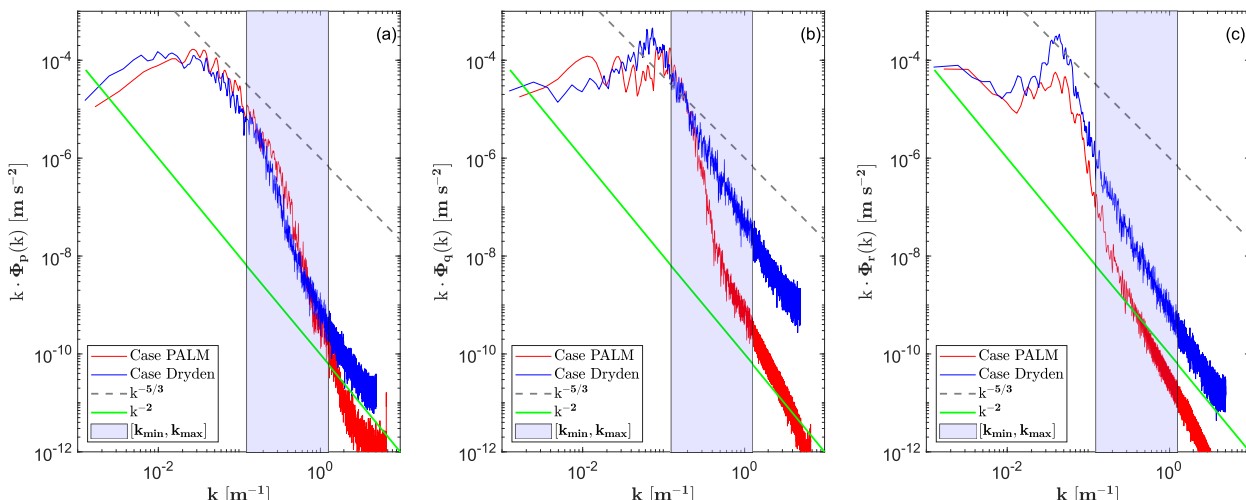

**Figure 17.** Spectra of airplane angular rate responses: (**a**) roll rate $p$, (**b**) pitch rate $q$ and (**c**) yaw rate $r$, derived from flight simulation output data of scenario Piz Corvatsch at an average altitude of 7500 ft (2280 m). The grey dashed line represents the reference $-3/5$-slope according to Kolmogorov's law [71]. The $-2$-slope is plotted in green. The box in light blue shading indicates the time frame of interest for the flight simulation experiment, which is determined by the lower limit wave number $k_{min}$ of $2\pi/50$ m$^{-1}$ and the high limit wave number $k_{max}$ of $2\pi/5$ m$^{-1}$.

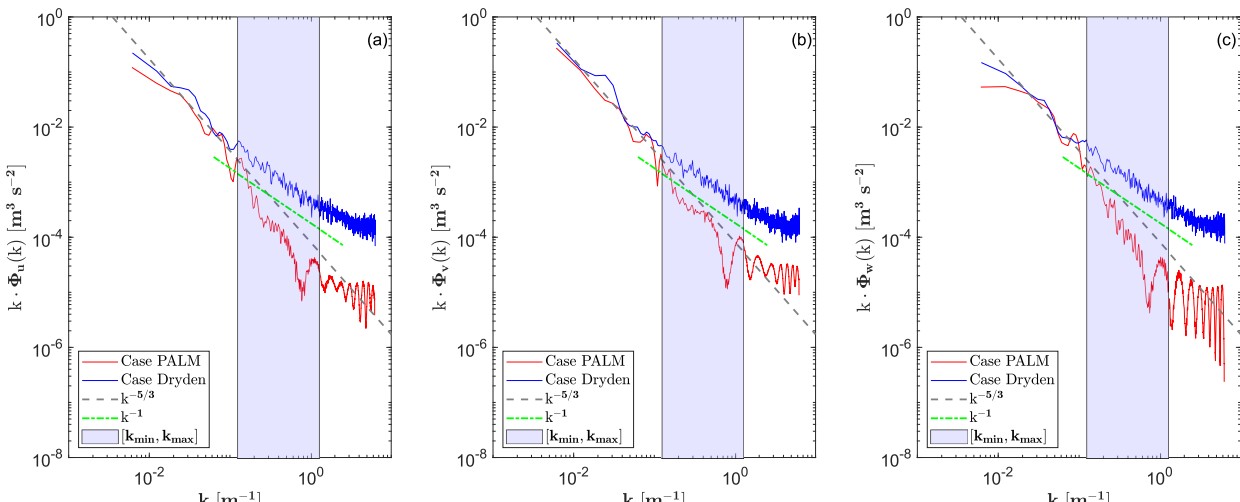

**Figure 18.** Spectra of wind velocity component (**a**) $u$, (**b**) $v$ and (**c**) $w$, derived from flight simulation output data of scenario Passo del Bernina at an average altitude of 10,500 ft (3200 m). The grey dashed line represents the reference $-3/5$-slope according to Kolmogorov's law [71]. The dashed dot green line denotes the reference $-1$-slope. The box in light blue shading indicates the time frame of interest for the flight simulation experiment, which is determined by the lower limit wave number $k_{min}$ of $2\pi/50$ m$^{-1}$ and the high limit wave number $k_{max}$ of $2\pi/5$ m$^{-1}$.

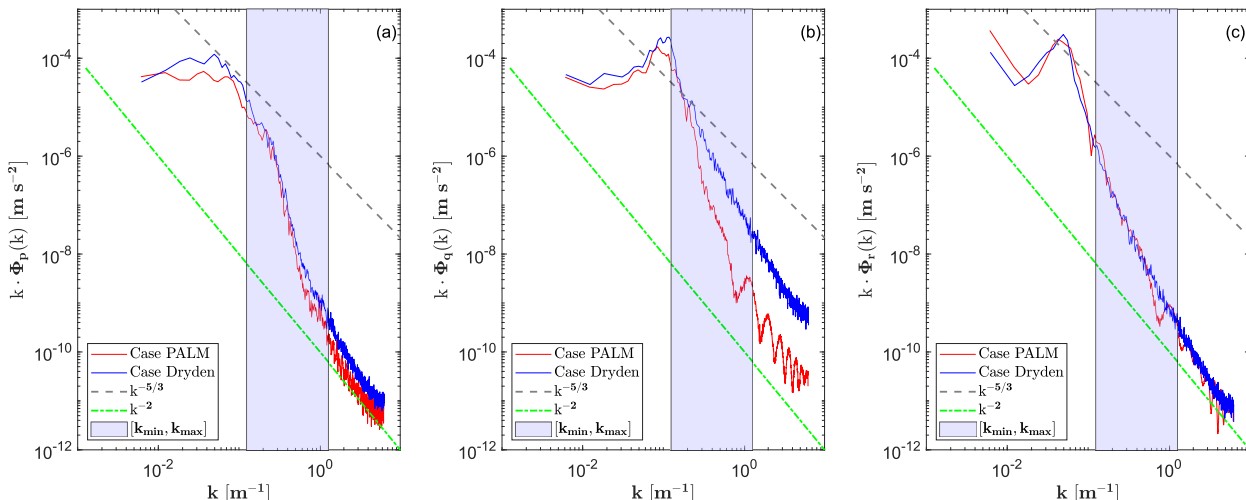

**Figure 19.** Spectra of airplane angular rate responses: (**a**) roll rate $p$, (**b**) pitch rate $q$ and (**c**) yaw rate $r$, derived from flight simulation output data of scenario Passo del Bernina at an average altitude of 10,500 ft (3200 m). The grey dashed line represents the reference $-3/5$-slope according to Kolmogorov's law [71]. The dashed green line denotes the reference $-1$-slope. The box in light blue shading indicates the time frame of interest for the flight simulation experiment, which is determined by the lower limit wave number $k_{min}$ of $2\pi/50$ m$^{-1}$ and the high limit wave number $k_{max}$ of $2\pi/5$ m$^{-1}$.

### 5.2.2. Results of Pilot Questionnaire

In this subsection, we summarize the pilots' evaluation of the atmospheric disturbance model and present an initial statistical analysis by using the question sheet presented in Appendix A.3. Here, the essential aim of the piloted simulation test was to answer the primary question of whether pilots can notice the differences between the two turbulence cases, rather than assess the absolute realism of the flight on the selected day and to review the causes of the accident. In fact, the validity and performance of other components in the flight simulation test environment, as well as the pilots' experiences flying under the presented test condition, may influence an assessment of the absolute reality. The evaluation of the flight simulation model is beyond the scope of this article.

In general, the pilots rated both turbulence cases between "fair" and "good". However, they were able to distinguish between the two approaches in turbulence characteristics and turbulence/task combination. The main criticism of the case Dryden was the presence of high-frequency noise. Due to the oscillations of the airplane angular rate responses, it was difficult to perform the flight tasks required. The pilots had to make considerable corrections and compensations. Pilot D, the experienced meteorologist, remarked that the high-frequency turbulence would be unrealistic on such a "normal" summer day. In terms of high-frequency turbulence, case PALM revealed an inverse image. On case PALM, the pilots commented that the high-frequency eddies were significantly less noticeable than expected. Due to this, it was much easier to complete the missions in case PALM than in case Dryden. In fact, the pilots' feedback is consistent with the energy spectra depicted in the previous section.

One positive characteristic of both cases, particularly case PALM, was the behaviour of low-frequency disturbances. The drifting in the mean wind was considered very realistic. This feedback from the pilots confirmed the correctness of the mean wind prediction in the LES simulations in turn. Differently from the conventional flight simulators, where the horizontal wind speed and direction on each flight are set to constant values, the mean wind fields in our flight simulation environment were derived from the LES simulations taking into consideration of their temporal and spatial evolution. Furthermore, the PALM

simulations used in this study took data from a mesoscale weather model as boundary conditions, which ensured the realistic initial conditions of the LES simulations.

We followed the approach described in [7] to statistically differentiate the two turbulence cases. The answers to the questionnaire are shown in Appendix A.5. The plots in bar charts can be interpreted as follows. For each question and turbulence case, we calculated 95% confidence limits for a true mean value of the pilots' ratings, e.g., by assigning the value 0 to "no comment", 1.0 to "not enough", 2.0 to "about right" and 3.0 to "too much". The red error bar denotes the 95% confidence interval for a given turbulence case. We can be 95% confident that the pilots can differentiate the two models, when both of the following requirements are satisfied at the same time:

(i)　the mean value of case PALM must lie outside the 95% confidence interval of case Dryden;

(ii)　the mean value of case Dryden must lie outside the 95% confidence of case PALM.

Figure A3 shows the pilots' ratings of turbulence intensity. This is an example of the pilots being able to distinguish between the two turbulence cases. The evaluations of the pilots were consistent. The case Dryden appeared to underestimate the turbulence intensities, while turbulence generated in the case PALM was more severe. Pilot A added further remarks to the comparison, stating that it was difficult to assess turbulence intensities without full motion feedback, especially when dealing with heave and side force. According to the pilot's expertise, turbulence should be classified as severe despite the fact that almost no aileron input was required to adjust for the roll of the aircraft.

The pilots' estimates for the turbulence realism are plotted in Figure A4. This is an example where the pilots can not differentiate the two turbulence cases statistically. Three of the four pilots rated the both cases "fair" or "good", while pilot C's rating for the case PALM was the lowest, being "poor". Pilot C consistently complained of too little disturbances felt during the flight experiment. The diverse feedbacks on the same model might be explained by the different recent flight experience. For the most part, pilot C has been flying smaller, lighter aircraft to complete his training.

Figures A5–A7 show the pilots' answers to the correctness of relative amplitudes of disturbances. All the pilots tended to think that the amplitudes of roll in both case were slightly lower than expected, as shown by the roll rate energy spectrum in Figure 19. For pitch, pilot A and pilot D rated both case "about right", while pilot B and pilot D were able to distinguish the two cases, but gave opposite opinions. Pilot B rated case Dryden "too much" and case PALM "about right". Pilot B felt the pitch oscillations in case PALM at approximately 10,500 ft high and added the comment that it was difficult for pilot to compensate. After analyzing the recorded flight routines (not shown), it was determined that pilot B completed only one complete circle at this altitude, whereas pilot D was asked to perform several level turns with various bank angles at this altitude, the majority of which occurred outside the child domain. The impression on the spatial discontinuity between parent and child domain could be less definite. For yaw, pilot A and pilot C rated differently, while the other two pilots rated both models "about right". It would be necessary to invite a larger group of pilots to conduct the test.

Figure A8 shows the pilots' evaluation of the patchy characteristics. Most of the pilots rated the both cases "about right". Pilot A thought turbulence bursts occurred in case PALM almost too abruptly and expected a bit more of fade-in and fade-out, even if the transition phase would last less than a second.

Pilots' ratings of low and high-frequency content are shown in Figure A10. As mentioned before, the pilots were able to differentiate the two cases in predicting high-frequency turbulence. Both cases need to be improved. One rapid solution would be activating the Dryden model in case PALM or the weighting factor o the Dryden model. Another way is to use a finer grid resolution, in order to shift the cut-off length of the LES simulations to the right side of the wave number area of interest. The plot in Figure A11 proves the validity of the low-frequency turbulence, despite Pilot B's rating of PALM as "not enough" and Pilot A's rating of Dryden case as "not enough".

The pilots rated the turbulence/task combination using the Cooper-Harper rating scale Figure A11; Figure A12 present the results of the evaluation. Pilots' ratings of the climb task show no difference between the two cases, while the pilots rated the level turn tasktask level turn in case PALM one full point lower than in case Dryden. This corresponds to the difference of in predicting high-frequency turbulence. Consequently, it was easier for the pilots to perform required tasks in case PALM than in case Dryden. The lower task climb ratings indicate that the climb manoeuvre was generally less challenging than the level turn on the test day. It would be interesting to carry out further piloted flight simulation test under ohter weather conditions, such as with a dominant vertical speed.

## 6. Conclusions and Outlook

This paper has introduced a method for generating atmospheric disturbances in a real-time simulation environment, by combining the statistical Dryden wind turbulence model with numerical wind fields data generated by the LES model PALM. The piloted simulation study was a scenario-based investigation focusing on an enhancement of flight safety over mountainous region.

PALM is a well-established model system, which has been widely applied for simulations of urban climate, atmospheric and ocean boundary layers. It provides a mesoscale nesting interface which can force the reanalysis data of a regional weather model such as COSMO with a spatial resolution $\mathcal{O}(\sim 1 \text{ km})$ as initial and boundary conditions onto the PALM simulation domain boundaries. The feature allowed us to simulate realistic mean wind fields in the flight simulator. Our PALM simulation domain with a grid resolution of 32 m was located mainly in the region of Samedan in Switzerland, including a small part within Italy on the south side of Piz Bernina (4049 m). To simulate the areas of most interest, we used an online LES-LES nesting system to embed two child domains with an equidistant grid spacing of 8 m in the parent domain. For the LES simulation, we selected a warm summer day with a suitable weather condition for performing mountain flights. The atmospheric boundary layer was stable and the horizontal winds were moderate on the test day. To evaluate the correctness of the LES simulations for predicting mean wind fields, we used measured wind data from a SwissMetNet weather station at Piz Corvatsch and a IMIS weather station at Passo del Bernina. The simulated data show acceptable agreement with the measured data.

We built a flight simulation test environment based on an existing Piper PA-28 flight simulation model. The atmospheric turbulence disturbance can be generated in two ways. The first one is using the PALM instantaneous data with high temporal resolution and local mesh refinement. The other one is using the hourly averaged data derived from LES simulation to simulate mean wind field, while the fluctuations are generated by Dryden turbulence model. We designed a flight simulation experiment to test the realism of both cases, in terms of turbulence characteristics and turbulence/task combinations. We carried out a statistical analysis to quantify the difference between the two models.

Based on the reported results, we can conclude that the two approaches can generate a realistic simulation environment for scenario-based investigation or pilot training on the subject of flight safety over complex terrain, although the high-frequency turbulence part needs to be improved further. Additional piloted simulation tests will be carried out for more difficult meteorological situations in order to determine the optimum weighting of the two models under various scenarios. Furthermore, we could implement the concept in a motion-based simulator, such as the other flight simulator at our centre's flight simulator for rotorcraft researchour centre for rotorcraft research. From a long term point of view, the urban air mobility industry may profit from the capability of such LES models like PALM for simulating urban climate, not only during the preliminary design phase, but also throughout certification and operating procedures, such as collision risk assessment and traffic control management, as propeller downwash and its interaction with obstacles would have significant impacts on the overall flight safety in low-level airspace.

**Author Contributions:** Conceptualization, X.L.; methodology, X.L.; software, X.L. and A.A.; validation, X.L., P.C. and L.M.; formal analysis, X.L.; resources, P.C. and L.M.; data curation, X.L. and A.A.; writing—original draft preparation, X.L.; writing—review and editing, X.L., L.M. and Y.F.; supervision, Y.F.; project administration, X.L. and L.M.; funding acquisition, X.L. and L.M. All authors have read and agreed to the published version of the manuscript.

**Funding:** This research was funded by the Federal Office of Civil Aviation in Switzerland grant number SFLV 2018-052.

**Institutional Review Board Statement:** Not applicable.

**Informed Consent Statement:** Not applicable.

**Data Availability Statement:** The data presented in this study are available on request from the corresponding author. The data are not publicly available due to the large data size and the specific data formats.

**Acknowledgments:** High Performance Computing (HPC) resources were provided by the Centre for Aviation (ZAV) at the Zurich University of Applied Sciences (ZHAW).

**Conflicts of Interest:** The authors declare no conflict of interest.

## Abbreviations

The following abbreviations are used in this manuscript:

| | |
|---|---|
| ABL | Atmospheric Boundary Layer |
| AMSL | Above Mean Sea Level |
| BEH | Passo del Bernina |
| COV | Piz Corvatsch |
| COSMO | Consortium for Small-scale Modeling |
| CFD | Computational Fluid Dynamics |
| CLS | Control Loading System |
| COV | Piz Corvatsch |
| DNS | Direct Numerical Simulation |
| FEAST | Finite Element Aircraft Simulation of Turbulence |
| FFT | Fast Fourier Transform |
| GUI | Graphic User Interface |
| IMIS | Intercantonal Measurement and Information System |
| INIFOR | Initialization and Forcing |
| LES | Large Eddy Simulations |
| LiDAR | Light Detection And Ranging |
| NetCDF | Network Common Data |
| PALM | Parallelized Large-Eddy Simulation |
| PPL | Private Pilot License |
| QGIS | Quantum Geographic Information System |
| RANS | Reynolds-averaged Navier–Stokes |
| SOBERT | Rotor Blade Element Turbulence |
| SLEVE | Smooth LEvel VErtical |
| SMN | SwissMetNet |
| SRTM | Shuttle Radar Topography Mission |
| STSB | Swiss Transportation Safety Investigation Board |
| SGS | Sub-grid scale |
| TKE | Turbulence Kinetic Energy |
| UTC | Coordinated Universal Time |
| WSL | Swiss Federal Institute for Forest, Snow and Landscape Research |
| ZAV | Centre for Aviation |
| ZHAW | Zurich University of Applied Sciences |

## Appendix A. Flight Simulation Test Campaign

*Appendix A.1. Weight and Balance*

**Table A1.** Weight and balance.

|  | MASS | | ARM | | |
|---|---|---|---|---|---|
|  | **Jb.** | **kg** | **in** | **m** | |
| Empty Mass | 1500 | 680.4 | 85.9 | 2.18 | |
| Position 1 | 176.4 | 80 | 80.5 | 2.04 | |
| Position 2 | 110.2 | 50 | 80.5 | 2.04 | |
| Position 3 | 110.2 | 50 | 118.1 | 3.00 | |
| Position 4 | 110.2 | 50 | 142.8 | 3.63 | |
| Fuel | 176.5 | 80 | 95.0 | 2.41 | |
| **Overall** | **2183.5** | **990.4** | **90.6** | **2.30** | 18.64% M.A.C. |

*Appendix A.2. Flight Test Card*

**Figure A1.** Flight test card.

*Appendix A.3. Question Sheet*

**ReDSim flight simulation test**

Run number: ______________

Pilot: ______________

Date: ______________

1. **Turbulence intensity**
   ◯ Light　◯ Moderate　◯ Servere　◯ Extreme
2. **Realism of turbulence**
   ◯ Very good　◯ Good　◯ Fair　◯ Poor　◯ Very poor
3. **Correctness of relative amplitudes of disturbances**

   |       | Not enough | About right | Too much | No comment |
   |-------|:----------:|:-----------:|:--------:|:----------:|
   | Roll  |     ◯      |      ◯      |    ◯    |     ◯      |
   | Pitch |     ◯      |      ◯      |    ◯    |     ◯      |
   | Yaw   |     ◯      |      ◯      |    ◯    |     ◯      |

4. **Patchy characteristics (variation of intensity-burst)**
   ◯ Much too continuous (monotonous)
   ◯ A little too continuous (monotonous)
   ◯ A little too patchy (monotonous)
   ◯ No comment
5. **Frequency content of turbulence**

   |                  | Not enough | About right | Too much | No comment |
   |------------------|:----------:|:-----------:|:--------:|:----------:|
   | Low frequencies  |     ◯      |      ◯      |    ◯    |     ◯      |
   | High frequencies |     ◯      |      ◯      |    ◯    |     ◯      |

6. **Other comments about realism**

   ................................................................................................................................

   ................................................................................................................................

   ................................................................................................................................

7. **Cooper rating (estimate of work load to perform the following tasks**

   | Level turn   | ◯ 1 | ◯ 2 | ◯ 3 | ◯ 4 | ◯ 5  |
   |--------------|-----|-----|-----|-----|------|
   |              | ◯ 6 | ◯ 7 | ◯ 8 | ◯ 9 | ◯ 10 |
   | Steady climb | ◯ 1 | ◯ 2 | ◯ 3 | ◯ 4 | ◯ 5  |
   |              | ◯ 6 | ◯ 7 | ◯ 8 | ◯ 9 | ◯ 10 |

8. **Other comments about work load**

   ................................................................................................................................

   ................................................................................................................................

   ................................................................................................................................

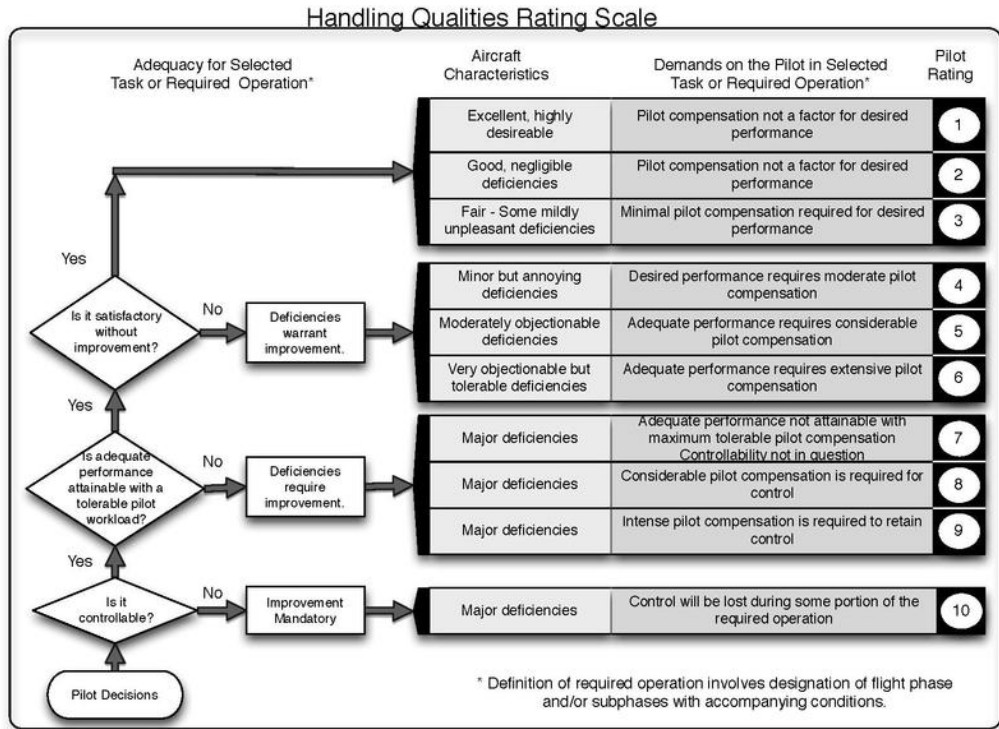

**Figure A2.** Cooper-Harper [69] handling qualities rating scale.

*Appendix A.4. Pilot's Experience*

**Table A2.** Pilots' experience.

| Pilot | Total No. of Hours | Hours under Similar Conditions | Technical Background |
|---|---|---|---|
| Pilot A | 95 | 10 | Aeronautical engineer |
| Pilot B | 175 | 25 | Aeronautical engineer |
| Pilot C | 55 | 5 | Aeronautical engineer |
| Pilot D | 155 | 20 | Meteorologist |

*Appendix A.5. Questionnaire Results*

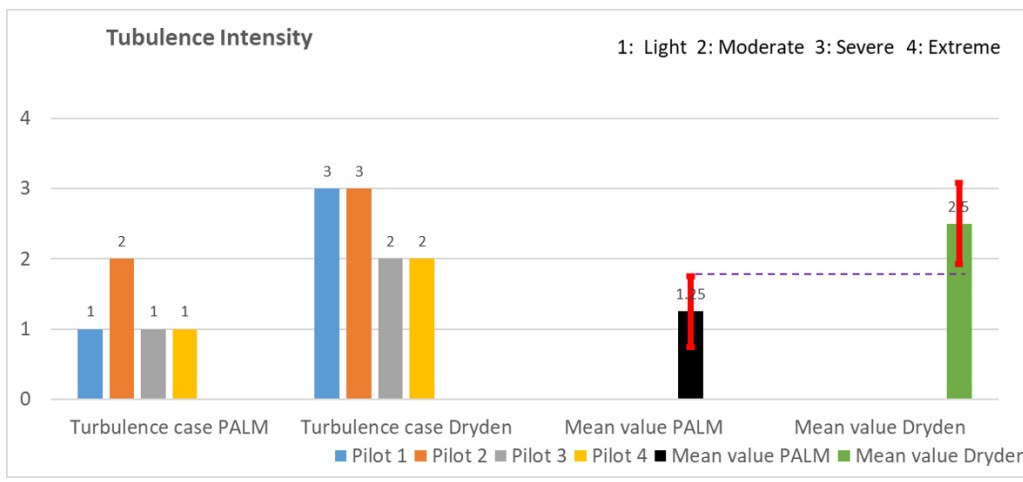

**Figure A3.** Turbulence intensity.

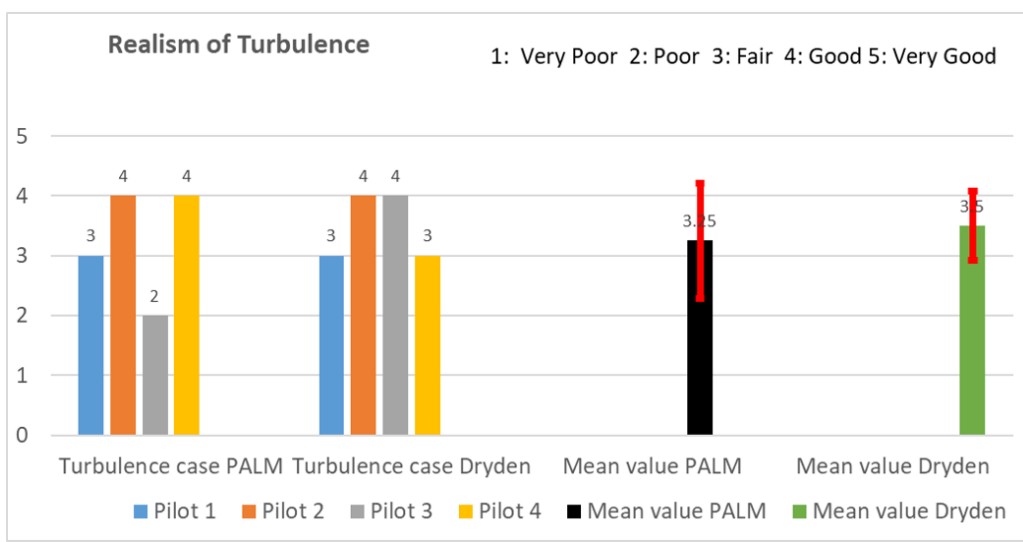

**Figure A4.** Realism of turbulence.

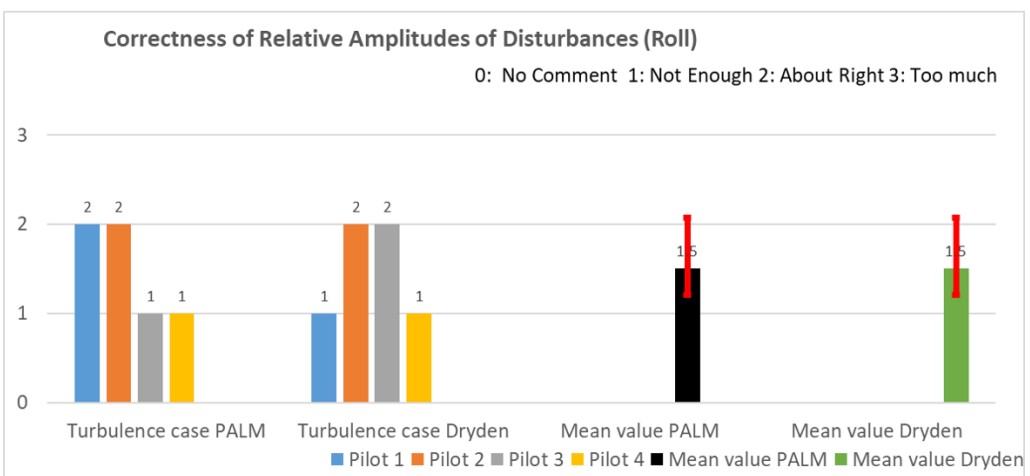

**Figure A5.** Correctness of relative amplitudes of disturbances (roll).

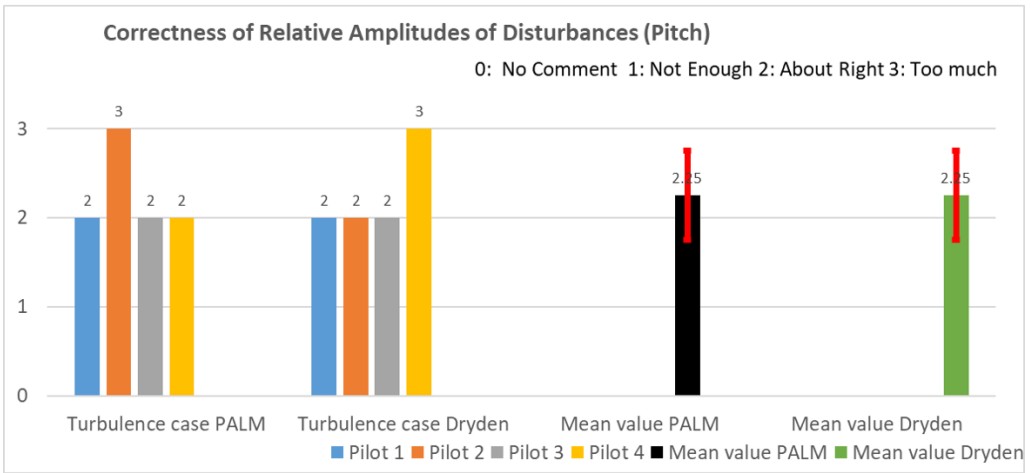

**Figure A6.** Correctness of relative amplitudes of disturbances (pitch).

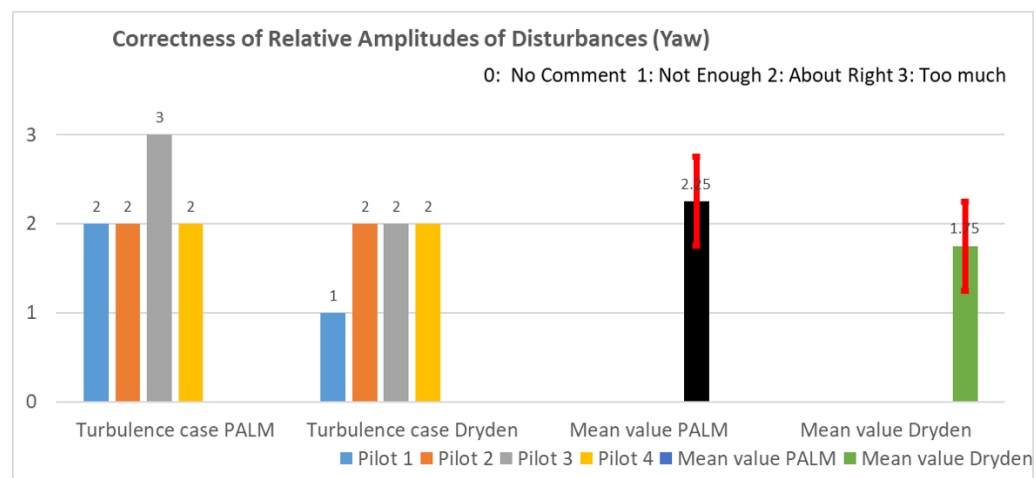

**Figure A7.** Correctness of relative amplitudes of disturbances (yaw).

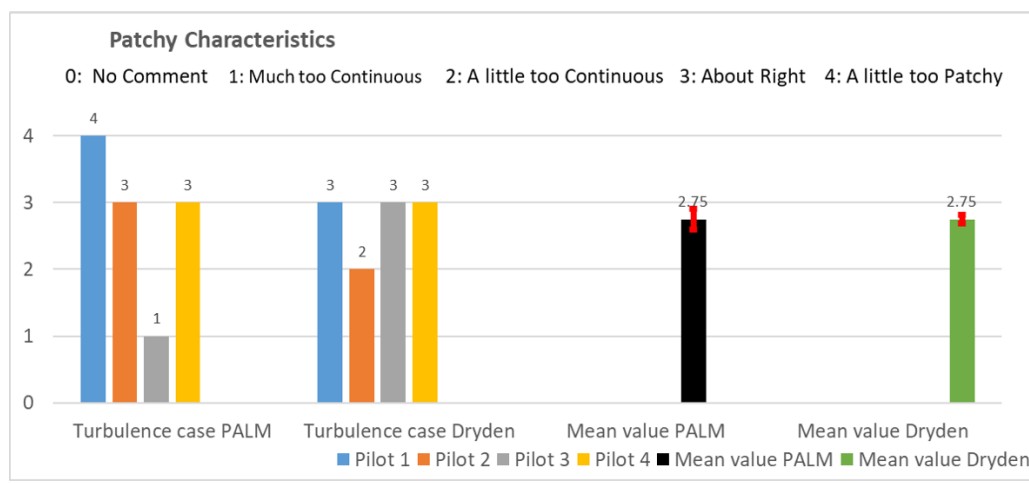

**Figure A8.** Patchy characteristics.

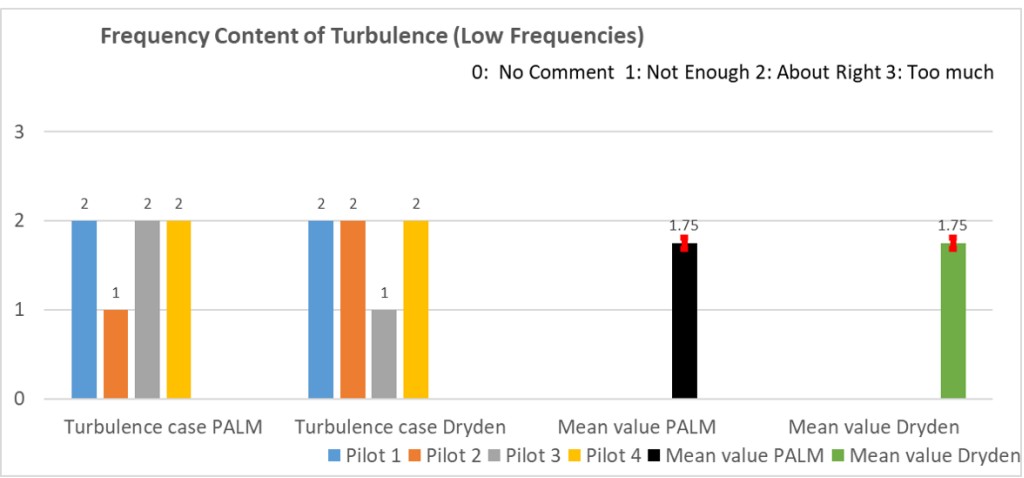

**Figure A9.** Frequency content of turbulence (low frequencies).

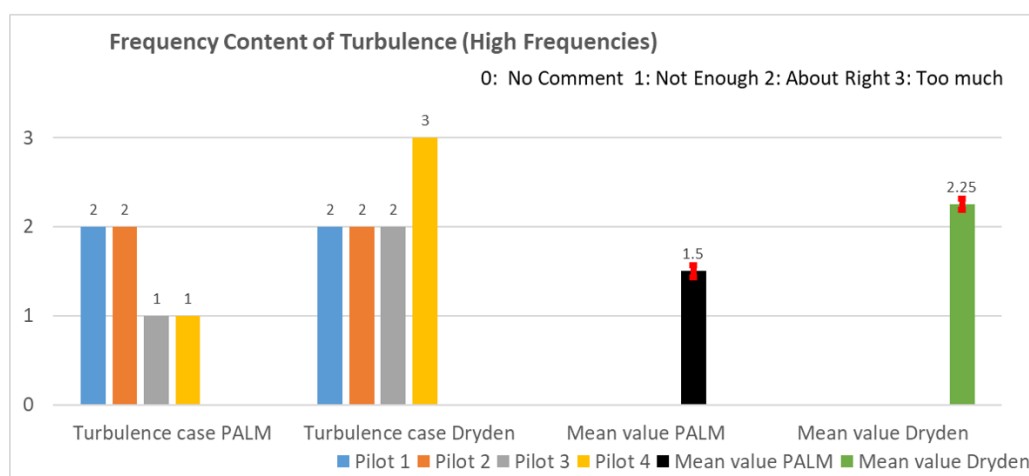

**Figure A10.** Frequency content of turbulence (high frequencies).

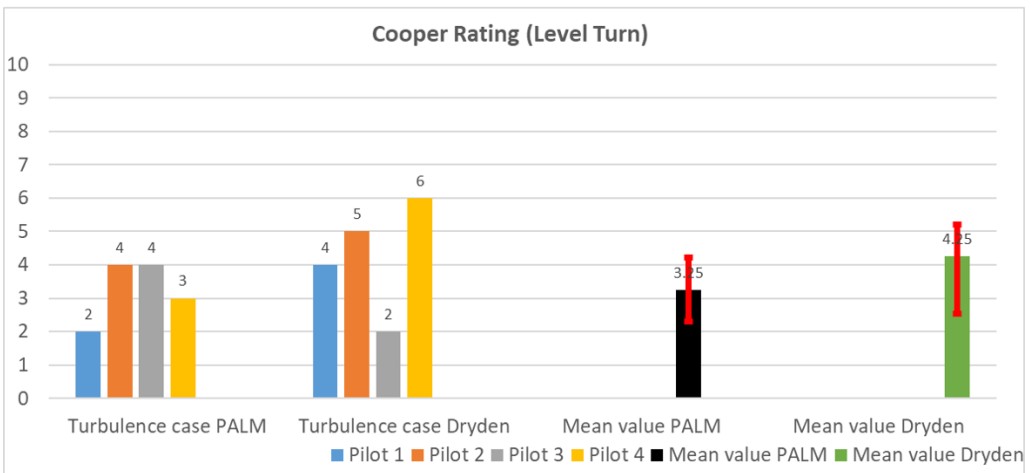

**Figure A11.** Cooper rating of task level turn.

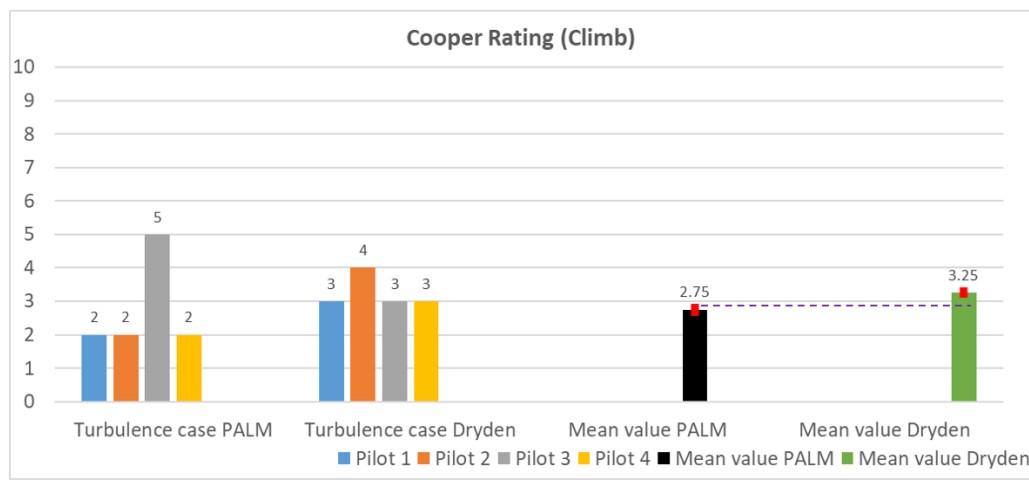

**Figure A12.** Cooper rating of task climb.

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
