# Peer review of "Atmospheric Disturbance Modelling for a Piloted Flight Simulation Study of Airplane Safety Envelope over Complex Terrain"

_aerospace, doi:10.3390/aerospace9020103_

Round 1

Reviewer 1 Report

This study presents verification of the LES-implemented flight simulator to generate more realistic atmospheric turbulence combining the Parallelized LES (PALM) flow field and the Dryden statistic model. The verification approaches were well designed with appropriate cases in two types of terrains. In addition, the statistical feedback from actual pilots through the questionnaire process was a good strategy to validate the authors’ model. Overall, the manuscript is well written and understandable but there are too many typos, which have to be corrected before publication. The following are some I found but authors have to check the whole manuscript again to correct all typos. And be aware if the following revisions are correct as authors intended.

  1. Line 2; “… envelope was was was developed…” -> “… envelope was developed…”
  2. Line 91; ω is used as the dimensional radian frequency here but ω is used again as the temporal frequency later, e.g. in Eq. (1). Not to make readers confused, the one of both symbols has to be replaced.
  3. Line 94; “… nose …” -> “… noise …”
  4. First paragraph in Page 7; “In today’s applications …” -> what is the meaning of “today”?
  5. Line 484; “… with forcus on he …” -> “… with forcus on the …”
  6. Caption of Figure 12; “The red scattered points solid lines present…” -> “The red scattered points present…”
  7. Line 569; authors mentioned that “the simulated data indicate a good agreement in the hourly averaged wind direction” in Figure 12(b). Then it’s better to show the additional line for the averaged values in the plot.
  8. Figure 13: a unit of the color bar is missing.
  9. Line 639; “… following the -5/3 law -5/3-law in …” -> “… following the -5/3-law in …”
  10. Line 640; “… show a different shape the energy is …” -> “… show a different shape; the energy is …”
  11. Line 646; “… angler rate p, q during …” -> “… angler rate p and q during …”
  12. Figures 16 – 19; Grey and green dashed lines are not easy to distinguish for readers even with the colored hard copy. Use different types of lines, e.g. grey dashed line and green dashed-dot line.
  13. Line 674; “… we summarized the pilots’ on the atmospheric …” -> “… we summarized the pilots’ estimates on the atmospheric …”
  14. Line 691; “… image.On example …” -> “… image. On example …”
  15. Line 724; “The pilots’ Estimates …” -> “The pilots’ estimates …”
  16. Line 764; “… manoeuvre …” -> “… maneuver …”
  17. Line 766; “… weather conditions , such as …” -> “… weather conditions, such as …”
  18. Line 768; “… a method by combing …” -> “… a method by combining …”

Author Response

Dear Reviewer

Thanks for your review and the comments.

Hereby, we provide a point-by-point response to your comments:

  1. Deleted “was was”
  2. The first one in the section 1 was replaced.
  3. Corrected
  4. “today’” replaced by “currently widely used”
  5. Corrected
  6. Corrected
  7. In the revised version, we specified the two hourly averaged values from simulation and measurement at the station BEH. It was already a discussion topic within our team between aerospace and meteorology researchers, as we had a different definition of the expression “good agreement”. The 10-degree deviation would be a perfect comparison result in meteorology or generally in geoscience, whereas the deviation could be rated only as “sufficient” in aerospace engineering. Due to that, we deleted the exact values in our manuscript. Further, the consistent temporal evaluation of wind direction is more important than the absolute values themselves in our application. Since addition lines or values in Figure 12(b) would make the plot confusing for the readers, so we modified the text directly. We hope the modification like this could be fine for you.
  8. Added the unit of “[m/s]”
  9. Deleted “-5/3 law”
  10. Added “;”
  11. “,” replaced by “and”
  12. Changed the line style of the green reference slope
  13. Corrected
  14. Added space
  15. Corrected
  16. “manoeuvre” is a spelling used in UK. In our University in Switzerland, we use officially UK English, like the name of our institute “Centre for Aviation”. Due to that, we changed the US spelling in the line 469 into the UK one, so that the word is consistently written in UK version in the whole paper.
  17. Deleted space
  18. Corrected

Furthermore, we have corrected other typos, rephrased some sentences, and added more clear explanations. We made some following modifications on the figure labels or figure style/resolution without using “markup” function of “changes” package in latex, since the package is not compatible with \autoref function for the cross citation in the text:

  • Figure 4: changed “(a)” to “(b)” in the last sentence;
  • Figure 6: added “….the fluid cells of …” in the third line of the label;
  • Figure 11: Passo del Bernina (BEH) instead of BEN;
  • Figure 12: deleted “solid lines”;
  • Figure 14:  changed “position 1” to “position 2” in the second line;
  • Figure 14-19: replaced by high-resolution vector images;
  • Figure 16-19: changed the line style of the green reference slope.

In the end, we completed the information of pilots’ experience in the appendix.

Best regards

Xinying Liu

Reviewer 2 Report

The manuscript describes a new model for the flight simulation environment and its implementation. The model adopts a Large Eddy Simulation (LES) approach to simulate the atmospheric disturbances for the flight simulation environment with a focus on the microburst events encountered in the atmosphere.

In general, the work is interesting. It offers insights how the high fidelity numerical model can be adopted to provide a more realistic input to real world applications. I would like to recommend the publication after the authors clarify the following issues.

  1. Line 2, Abstract - "was was was" -> "was"
  2. The authors claim that PALM model is a finite difference flow solver. However, in Section 3.3.2 (Eq. 62), the method adopts the flux concept to approximate the advective terms in the governing equations. The flux concept is actually a finite volume concept, rather than finite difference concept. For finite difference, the partial derivatives are all directly approximated by difference operator without requiring any flux evaluation. Is PALM really a finite difference scheme? Could the authors explain more about this?
  3. In the figure showing the results, e.g. Figure 7, what are the lines denoted by 1h, 2h etc? Are they the hourly averaged of the first hour, second hour etc? Please clarify them in the manuscript.
  4. In Figure 12b, should an hourly averaged value be shown to support the claim that "measured and simulated data indicate a good agreement in the hourly averaged wing direction in the selected time range" (Line 569-570)?

Author Response

Dear Reviewer

Thanks for your review and the comments.

Hereby, we provide a point-by-point response to your comments.

  1. Corrected
  2. Yes, the domain in PALM is really discretized in space using finite differences. PALM uses the Arakawa staggered C-grid, in which the scalar values locate in the cell centre while vector variables (e.g. u, v and w) defined in the face centre (see figure 1). It means the fluxes stagger half a grid length related to the advected quantity. Due to this, the flux advection adopts the scheme presented in eq. 62, which looks similar with the FV-scheme of the solvers to calculate wall-bounded flow in our aeronautical application. This scheme is widely used in the environment research. The flux in eq. 62 has been described explicitly in the revised version. For detailed information about the scheme, refer to reference 62.
  3. Yes, they are the hourly averaged vertical profiles. The averaging interval for output of vertical profiles was set to 600 s. It has been clarified at the beginning in the section 5.1.1 in the revised version.
  4. In the revised version, we have specified the two hourly averaged values from simulation and measurement at the station BEH. It was already a discussion topic within our team between aerospace and meteorology researchers, as we had a different definition of “good agreement”. The 10-degree deviation would be a perfect comparison in meteorology or generally in geoscience, whereas the deviation could be rated only as “sufficient” in aerospace engineering. Due to that, we deleted the exact values in our manuscript. Further, the consistent temporal evaluation of wind direction is more important than the absolute values themselves in our application. Since addition lines or values in Figure 12(b) would make the plot look confusing for the readers, so we modified the text directly. We hope the modification like this could be fine for you.

Furthermore, we have corrected other typos, rephrased some sentences, and added more clear explanations. We made some following modifications on the figure label or figure style/resolution without using “markup” function of “changes” package in latex, since the package is not compatible with \autoref function for the cross citation in the text:

  • Figure 4: changed “(a)” to “(b)” in the last sentence of the label;
  • Figure 6: added “….the fluid cells of …” in the third line of the label;
  • Figure 11: Passo del Bernina (BEH) instead of BEN;
  • Figure 12: deleted “solid lines”;
  • Figure 14: changed “position 1” to “position 2” in the second line;
  • Figure 14-19: replaced by high-resolution vector images;
  • Figure 16-19: changed the line style of the green reference slope.

In the end, we completed the information of pilots’ experience in the appendix.

Best regards

Xinying Liu

Reviewer 3 Report

Comments on the paper entitled:  Atmospheric disturbance modelling for a piloted flight simulation study of airplane safety envelope over complex terrain.

The submited work can be of interest to Aerospace readers because it presents the application of a LES model (PALM) to produce turbulence in the ABL that can affect the safety of the real flights, especially in complex terrain. In the work, the modelling of non-homogeneous turbulent flow with a suitable subgrid parametrization is highlighted. Also, it shows the turbulent properties from profiles of wind, potential temperatures, turbulent fluxes of heat and momentum at different times of the simulated tests. In addition, the spectra of the wind components and the angular rate and their comparison with the Dryden model, showing consistency with the -3/5 Kolmogorov’s law, is remarkable. Several flight simulation tests have been carried out in delimited areas by expert pilots who have provided information about the aircraft response.

The paper, however, has a lack in the definition of the coordinates frame for the simulation tests over the selected zone and some mistakes in the text that must be corrected before being published in Aerospace.  Below I will list the aspects that need to be reviewed by the authors.

  1. The frame of coordinates must to be defined more clearly. It seems that is fixed for each test, but in the manuscript it does not say if the x-axis is parallel to any particular direction, e.g. W-E, or to another direction. On the other hand, I don’t know if z-coordinate (vertical) is measured from the sea level or from some point of the terrain. Clarify it in the description of the test campaign or before.
  2. Line 283, following to the previous comment, it says: “height range from about 2000 m to less than 5000 m”, but, where is it measured from?
  3. Line 410, it says “two child domains of 2048 m x 2048 m x 3584 m were nested in the parent domain”, but what is the height of the bottom of these boxes, from the ground or from sea level? This must be clarified as well.
  4. Several particular mistakes have been detected throughout the document:
    • Page 6, in Eq (9), second equation, for pg is wrong.
    • Page 7,  line non numerated between eq(18) and eq(19).  “The the…”, it is doubled.
    • Page 8, line 259+6, into Table 1. “hpa” must be “hPa”
    • Page 13, line 299+2. “artificiall”  is “artificial”
    • Page 14, eq (67). The symbol (!=), is it an error? What does it mean?
    • Page 14, line 304. “requring” is “requiring”
    • Page 16, line 367. What does “… spread over 80 terrain…” mean?  
    • Page 17, line 399. “anterpolation” is “interpolation”
    • Page 17, Figure caption Fig 4. Last line “The red frames in (a)…” must be “The red frames in (b)…”
    • Page 24. Figure caption of Fig 11. Second line “Passo del Bernina (BEN)… ” is “Passo del Bernina (BEH)…”
    • Page 26. Figure caption of Fig 14. 2nd line:  “(position 1)”, it is duplicated.
    • Page 27. Line 615 “Vtas … in the true airspeed”. What does “true airspeed” mean?, perhaps it is the average velocity around the aircraft? Or, the velocity of the unperturbed flow? Clear it up.
    • Page 27, line 617. “initial subrange”, are you referring to “inertial subrange”?
    • Page 27, line 619. Same that in the previous “interial” is “inertial”.

As I said above, the paper needs to be revised and all comments responded before it can be published.

Author Response

Dear Reviewer

Thanks for your review and the comments.

Hereby, we provide a point-by-point response to your comments.

  1. The calculation domain is fixed with x-axis in E-W direction and y-axis in N-S direction. A detailed explanation of the coordinates has been added in section 3.5 below table2.
  2. Above mean sea level (AMSL). It has been clarified in the text.
  3. The bottom of the child domains and the parent domain is determined by the lowest height of 1746 m (AMSL) in the selected test area. This number is obtained from the terrain model, digital elevation model (DEM). This has been clarified in section 3.5 below table (see point 1). Note that the red shadowed in Figure 6 represent the fluid cells of the child domains.
  4.  
    • Corrected
    • Deleted one “the”
    • Corrected
    • Corrected
    • In Eq67, the symbol is not an error. In general, it means “defined”. We used this notation to distinguish the definition from “normal” equality. In the text, it was written the first spatial derivative is to be defined as 0 to close the equation for solving the pressure disturbance. In the revised version, an explanation of this notation is included in the text, so that the mathematical procedure can be clear for readers.
    • Corrected
    • The definition is originally from reference 67. It means 80 levels in the terrain-following coordinate, which is used in the regional weather model COSMO.
    • It is not a typo. We meant indeed anterpolation, which is transpose of interpolation or we can also call it “adjoint of interpolation”. It’s a mathematical definition. Passing the values from child domain (fine mesh) to parent domain (coarse mesh) should be described as anterpolation. For details see reference 68.
    • Corrected
    • Added space
    • Corrected
    • Added “of the airplane” to avoid misunderstanding. The true airspeed (TAS; also KTAS, for knots true airspeed) of an aircraft is the speed of the aircraft relative to the air mass through which it is flying. The true airspeed is important information for accurate navigation of an aircraft. Traditionally it is measured using an analogue TAS indicator, but as the Global Positioning System has become available for civilian use, the importance of such analogue instruments has decreased. In the flight simulation model, normally we record also other airspeeds like, calibrated airspeed, equivalent airspeed and indicated airspeed. Here, we use the true airspeed for the calculation in the 5.2.1.
    • Corrected
    • Corrected

Furthermore, we have corrected other typos, rephrased some sentences, and added more clear explanations. We made some following modifications on the figure label or figure style/resolution without using “markup” function of “changes” package in latex, since the package is not yet compatible with \autoref function for the cross citation in the text:

  • Figure 4: changed “(a)” to “(b)” in the last sentence of the label;
  • Figure 6: added “….the fluid cells of …” in the third line of the label;
  • Figure 11: Passo del Bernina (BEH) instead of BEN;
  • Figure 12: deleted “solid lines”;
  • Figure 14: changed “position 1” to “position 2” in the second line;
  • Figure 14-19: replaced by high-resolution vector images;
  • Figure 16-19: changed the line style of the green reference slope.

In the end, we completed the information of pilots’ experience in the appendix.

Best regards

Xinying Liu

Round 2

Reviewer 3 Report

I have no new comments. The paper is fine after the first revision. In my opinion it can be published in Aerospace.